# Theoretical and Empirical Advances in Forest Pruning

Albert Dorador

University of Wisconsin - Madison

`albert.dorador@wisc.edu`

Regression forests have long delivered state-of-the-art accuracy, often outperforming regression trees and even neural networks, but they suffer from limited interpretability as ensemble methods. In this work, we revisit forest pruning, an approach that aims to have the best of both worlds: the accuracy of regression forests and the interpretability of regression trees. This pursuit, whose foundation lies at the core of random forest theory, has seen vast success in empirical studies. In this paper, we contribute theoretical results that support and qualify those empirical findings; namely, we prove the asymptotic advantage of a Lasso-pruned forest over its unpruned counterpart under weak assumptions, as well as high-probability finite-sample generalization bounds for regression forests pruned according to the main methods, which we then validate by way of simulation. Then, we test the accuracy of pruned regression forests against their unpruned counterparts on 19 different datasets (16 synthetic, 3 real). We find that in the vast majority of scenarios tested, there is at least one forest-pruning method that yields equal or better accuracy than the original full forest (in expectation), while just using a small fraction of the trees. We show that, in some cases, the reduction in the size of the forest is so dramatic that the resulting sub-forest can be meaningfully merged into a single tree, obtaining a level of interpretability that is qualitatively superior to that of the original regression forest, which remains a black box.

## 1. Introduction

The benefits of combining forecasts have been documented in the literature at least since Laplace [1], gaining traction after the seminal works of Reid [2, 3] and Bates and Granger [4].

Rather remarkably, the overall success of the ensemble does not rely on the strength of the individual base learners. In fact, the base learners are either assumed or even designed to be individually weak in some ensemble methods, e.g. Boosting [5], BART [6]. However, this does not mean that having individually strong base learners is necessarily detrimental to ensemble performance. In fact, the opposite has been proved to be true in some methods: in random forests [7], the generalization error of the forest depends positively on the strength of the individual trees in the forest, while the dependence is negative on the correlation among them.

This behavior is not exclusive to random forests, and has led to the emergence of an important line of research focused on devising methods to identify subsets of base learners within a given ensemble that lead to better out-of-sample performance compared to the full ensemble. Margineantu's pioneering work in this topic [8] studies ensemble pruning in the context of the AdaBoost model [9] with classification trees as base learners, empirically showing that ensemble size can sometimes be substantially reduced while maintaining performance. Similar results are presented in Zhou et al. [10] for neural networks, and in Reid and Grudic [11] in the context of stacking different families of base classifiers, including neural networks, support vector machines, k-nearest neighbors, decision trees and random forests. In the specific context of random forests, of notable mention are the works of Zhang and Wang [12] and Bernard et al. [13], which show the success of forest-pruning methods in a classification context through extensive simulations. In the present work, among other contributions, we extend those empirical findings to the regression context, obtaining extreme forest reductions of up to $99\%$ while improving out-of-sample performance, which we support with theoretical results.

The application of ensemble pruning to regression forests is particularly appealing, as this technique bridges the gap between the highly interpretable regression trees and the highly accurate (although uninterpretable) regression forests. Interpretability is a very desirable feature, especially in high-stakes scenarios [14].

Second Conference on Parsimony and Learning (CPAL 2025).

The rest of the paper is organized as follows: Section 2 reviews the forest pruning problem along with different proposed solutions. Section 3 provides a novel way to visualize the forest-pruning solution. Section 4 presents our theoretical contributions, including the essentially assumption-free asymptotic superiority of Lasso-pruned regression forests over their unpruned counterparts, as well as finite-sample generalization bounds for regression forests pruned by different means, including Lasso. Section 5 outlines the testing conditions used in our empirical tests with synthetic and real data, whose main results are shown in Section 6 and 7, respectively. In Section 8 we combine the trees in a pruned forest into a single tree for maximum interpretability, and compare this approach with traditional single tree pruning. Section 9 concludes.

While some of our contributions (e.g. generalization bounds) are specific to regression, some others can easily be adapted to classification (e.g. the Best Sub-Forest algorithm, or our method to visualize a pruned forest).

## 2. The forest pruning problem

### 2.1. Problem definition

Given a decision forest, one wishes to find a small sub-forest that maximizes accuracy, ideally outperforming the original full forest. This application of ensemble pruning to decision forests is known as "forest pruning". This notion bears similarities with that of tree pruning [15, 16], where some internal nodes of a given tree are removed; in the case at hand, entire trees are removed, instead.

In practice, after fitting a regression forest with our training data, we need to solve the following constrained integer program for $y \in \mathbb{R}^{n_v}$, where $n_v$ is the size of our validation set:

$$\min_{x_1, x_2, \ldots, x_B \in \{0,1\}} \left\| y - \frac{1}{\sum_{i=1}^{B} x_i} \sum_{i=1}^{B} \hat{t}_i x_i \right\|_2^2 \quad \text{subject to} \ \ 1 \leq \sum_{i=1}^{B} x_i \leq K(B) \tag{1}$$

where $\hat{t}_i \in \mathbb{R}^{n_v} \ \forall i \in \{1, 2, \ldots, B\}$ is the validation-set prediction of the $i$-th tree in the forest and $K(B) \leq B$ is the maximum number of trees one wishes to have in the final sub-forest, typically a number much smaller than $B$, the total number of trees in the original full forest.

Optimal sub-ensemble selection with $K(B) = B$ is NP-hard [17]. Several solutions have been proposed, and we review the main ones next. See Appendix A for the pseudo-code that implements each approach.

### 2.2. Sequential forward selection (SFS)

This heuristic, first introduced in Margineantu [8], is an analog of the well-known forward stepwise variable selection method used in regression, and therefore it shares the same advantages (fast and relatively successful) as well as disadvantages (may miss the global optimum due to its greedy selection order).

In this case, one begins with an empty set of selected trees and sequentially adds one more tree from the pool of trees in the full forest based on which new tree can improve most the validation mean squared prediction error (MSPE) considering the trees already selected in the same fashion. As there are $B$ trees in the original full regression forest, thus requiring a total of $\mathcal{O}(B^2)$ performance checks, in turn each requiring $\mathcal{O}(nB)$ operations (MSPE computation), the computational complexity of this algorithm is $\mathcal{O}(nB^3)$, where $n$ is the number of observations in the validation set.

### 2.3. Modified sequential backward selection (SBS')

This method is a straightforward variation of SFS where we start with the full forest and sequentially discard the tree whose removal from the forest has the least impact on prediction accuracy (in the validation set) until only one tree of the forest survives, then find the point in the tree sequence that minimized MSPE. The pseudo-code of our implementation, which adapts to the regression setting the original method described in Zhang and Wang [12] for classification tasks is in Appendix A. Given the similarity with SBS and the fact that SFS was shown superior in Bernard et al. [13], we test SBS' instead of the original SBS method in this work.

In turn, several variations of this idea have been proposed, including the 'winner method' in Zhang and Wang [12], denoted here by SBS', as well as a novel application of the Ghost variable idea [18] that we have adapted

and studied in the context of forest pruning. In Appendix B, we introduce this latter method and show it is in fact equivalent to the traditional SBS.

In all cases, the computational complexity analysis is identical to that of SFS, and hence $\mathcal{O}(nB^3)$ too.

## 2.4. The best sub-forest (BSF) method

The most intuitive approach to solving the forest pruning problem is a brute-force approach, where one checks the validation-set performance of each possible subset of trees of cardinality at most $K \leq B$, for $K \in \{1, \ldots, B\}$. However, this approach has computational complexity $\Theta(2^B)$.

Nevertheless, it is possible to bring the complexity down to pseudo-polynomial time (polynomial after fixing $K$, independent of $B$) by noticing the following simple observation, assuming $K \leq B/2$: $\sum_{j=1}^{K} \binom{B}{j} \leq K\binom{B}{K} \in \Theta(B^K)$. This is how our proposed method, BSF, works: solve the original forest pruning problem with $K < B/2$ fixed ex ante. Observe that, typically, only an extremely small subset of the power set of trees in the original forest is in fact checked by design, which eases the scalability concerns about this method. Our empirical results show that $K = 3$ (or even $K = 2$) might be enough in some cases (see Section 8).

## 2.5. The (non-negative) Lasso solution

The Lasso [19] solution to the forest pruning problem is in fact a very natural one, as it constitutes a convex relaxation of the original problem, which can then be optimally solved in a much more efficient manner. The application of Lasso to prune ensembles finds its roots in a technical report by Friedman and Popescu [20] where the authors take a closer look at ensemble learning from the perspective of high-dimensional numerical quadrature. Since then, this application of the Lasso framework has been explored in the literature in different contexts: classification forests [16], stacking in multi-class classification [11], or ensembles of neural networks for classification tasks [17]. In this paper, we revisit this idea in the context of regression forests and modify it by imposing non-negativity constraints.

Thus, for $y \in \mathbb{R}^{n_v}$ in the validation set, consider now the following optimization problem instead:

$$\min_{\beta_1, \beta_2, \ldots, \beta_B \geq 0} \left|\left| y - \sum_{i=1}^{B} \hat{t}_i \beta_i \right|\right|_2^2 + \lambda \sum_{i=1}^{B} \beta_i \tag{2}$$

where $\hat{t}_i \in \mathbb{R}^{n_v} \forall i \in \{1, 2, \ldots, B\}$ is the validation-set prediction of the $i$-th tree in the forest and $\lambda \geq 0$ is the $\ell_1$ penalty parameter (a tuning hyperparameter).

The benefits of including non-negativity constraints are diverse: potentially improved generalization (estimated coefficients will typically be upper bounded by 1 under i.i.d. sampling, avoiding additional 'compensatory' coefficients), weaker regularity conditions to achieve variable-selection consistency (as the absolute value operator in the Irrepresentable Condition [21] is no longer needed), and may improve interpretability (avoids artificial solutions where the sign of all predictions by one or more trees is reversed).

Under regularity conditions, non-negative Lasso solutions possess parameter estimation and variable-selection consistency [22]. In forest-pruning, these properties mean that, as $n \to \infty$ and under the corresponding regularity conditions, Lasso pruning will not only correctly identify the relevant trees in the forest, but also will be able to find their true weights (with high probability), instead of using uniform weights as the random forest and most other tree ensemble methods do (which, in general, are biased even as $n \to \infty$, since they are not estimated but determined ex-ante to be uniform).

Our algorithm makes use of the R library `glmnet` (version 4.1-7) [24]), with complexity $\mathcal{O}(nB^2 + B^3)$, the same as Ordinary Least Squares (OLS).

To further aid interpretability, we suggest imposing a limited maximum forest size. Then, if the regularized solution yields a number of trees that exceeds our desired threshold, we refit the Lasso model forcing the smallest coefficients exceeding our threshold to be zero, similarly to Wu et al. [22]. This refitting procedure, which relates Lasso post-processing to the Relaxed Lasso [25] takes place at most once and is partly motivated by the fact that the strategy of simply increasing $\lambda$ may not behave as expected, since the number of nonzero

$\beta_i$ coefficients is not monotonic in the magnitude of $\lambda$, as mentioned in the literature without formal proof [19]. We formally state and prove this result in Appendix D.

# 3. Visualizing the pruned forest

After the regression forest of cardinality $B$ has been fitted to the training data we can obtain the $n_v \times B$ matrix of predictions $\mathcal{P}_v$ in the validation set, where each column holds the predictions for each of the $n_v$ observations in the validation set made by a given tree in the forest.

One way to visualize the trees in the forest is to focus on two of their key properties, namely their accuracy and their correlation with the rest of trees in the forest. The first property can be directly obtained from $\mathcal{P}_v$, while a proxy for the second property is given by the $B \times B$ matrix of (sample) correlations $\mathcal{C}_v$ that can be derived from $\mathcal{P}_v$. An analog procedure can be followed for the test set.

Having computed the matrix of sample correlations $\mathcal{C}_v$, we can define an associated $B \times B$ distance matrix $\mathcal{D}_v$ as follows: $\mathcal{D}_{v_{ij}} = \sqrt{\frac{1}{2}\left(1 - \mathcal{C}_{v_{ij}}\right)} \in [0, 1]$. It is known that this mapping is indeed a metric [26].

Then, given such a distance matrix, we can obtain a 2-dimensional representation of the trees by Multidimensional Scaling (MDS). An MDS algorithm projects each object onto an $N$-dimensional space ($N = 2$ in this case) such that the between-object distances are preserved as well as possible. Below is an example that illustrates the BSF solution to the forest-pruning problem in the Diamonds dataset (available in the `ggplot2` R library). The two selected trees (in turquoise) are large and far apart, meaning that the BSF solution (more accurate than the full forest solution comprising 200 trees, see Figure 4) selected only two trees with high individual performance but relatively weak correlation.

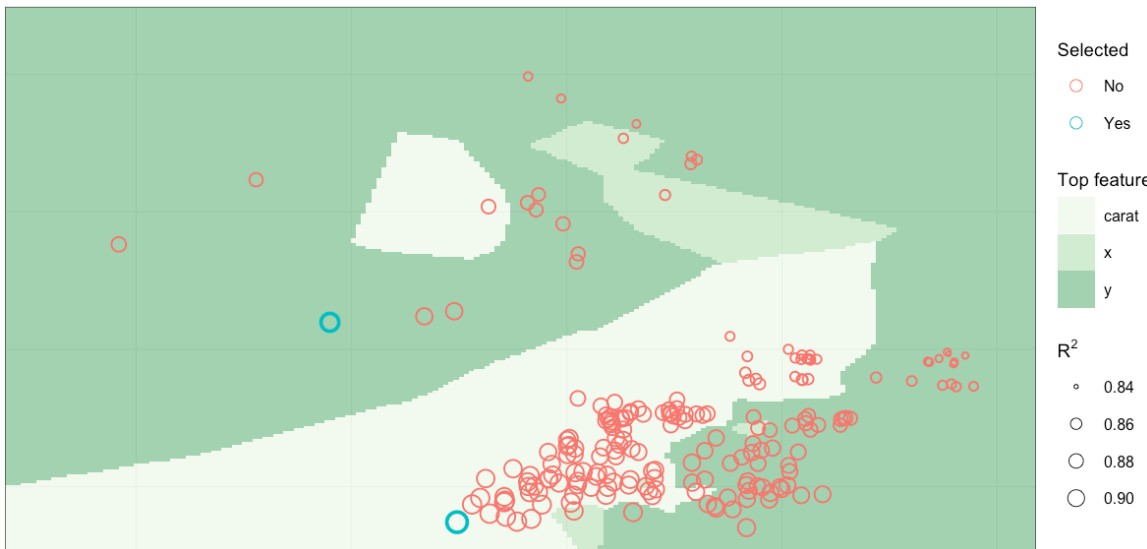

Figure 1: Ensemble of 200 CART trees pruned down to two by BSF (Diamonds dataset)

# 4. Theoretical results

In what follows, we restrict attention to bounded regression problems. We deem this simplifying assumption reasonable for most real-life applications, due to the finite nature of physical resources.

**Definition 4.1.** Let $\mathcal{Y} \subseteq \mathbb{R}$ be a label space. A regression problem is said to be bounded if the loss function $\ell$ involved is bounded, i.e. $\exists M > 0$ such that $\ell(y, y') \leq M$ for all $y, y' \in \mathcal{Y}$.

For squared losses, $M := [\sup(\mathcal{Y}) - \inf(\mathcal{Y})]^2$, assumed finite. However, this bound tends to be extremely conservative, as the cardinality of the feature space partition induced by any forest would not allow deviations of that caliber under mild assumptions e.g. Lipschitzness of the underlying regression function.

### 4.1. The asymptotic advantage of Lasso pruning

Given a trained forest consisting of $B$ trees and a dataset $S = \{(x_1, y_1), \ldots, (x_n, y_n)\} \equiv (X, Y) \in (\mathcal{X} \times \mathcal{Y})^n$ drawn i.i.d. from a distribution $\mathcal{D}$ over $\mathcal{X} \times \mathcal{Y}$, consider the linear regression model $Y = \tilde{X}\beta + \varepsilon$ where $\varepsilon$ has i.i.d. zero-mean $\sigma$-sub Gaussian components and $\tilde{X} = F(X)$ is the $n \times B$ matrix of predictions that results from applying each tree model in the given forest to the features $X$ observed (i.e. each of the $B$ trees in the forest contributes an $n$-dimensional vector of predictions). In this context, it makes sense to assume the elements of $\beta$ are non-negative, informing the eventual use of the non-negative variant of the Lasso. The same conclusions (with minor adaptations) hold in a more general ensemble pruning context, as the prediction function $F()$ above can be viewed more generally and not restricted to the case of a random forest.

Let $z = (y, \tilde{x}_1, \ldots, \tilde{x}_B)$ be an out-of-sample observation, where $\tilde{x}_1, \ldots, \tilde{x}_B$ are stochastic and possibly dependent random variables, such that, for some $M > 0$, $|\tilde{x}_j| \leq M$ holds almost surely for all $j$. As before, $y = \sum_{j=1}^{B} \beta_j \tilde{x}_j + \varepsilon$, where $\varepsilon$ is a zero-mean $\sigma$-sub Gaussian random variable. Now, let $z_i = (y_i, \tilde{x}_{i,1}, \ldots, \tilde{x}_{i,B})$ be i.i.d. copies of $z$, forming a training set $\tilde{S} = \{z_1, \ldots, z_n\} \equiv (Y, \tilde{X})$.

**Theorem 4.2.** *(Out-of-sample risk bound of non-negative Lasso) With the previous (essentially assumption-free) setup,*

$$\mathbb{E}[(\tilde{x}^T\beta - \tilde{x}^T\hat{\beta})^2] \leq 2\tau M\sigma\sqrt{\frac{2\log(2B)}{n}} + 8\tau^2 M^2\sqrt{\frac{2\log(2B^2)}{n}} \tag{3}$$

*where $\hat{\beta}$ satisfies*

$$\hat{\beta} \in \underset{\tilde{\beta}_1, \ldots, \tilde{\beta}_B \geq 0}{\arg\min} \ ||Y - \tilde{X}\tilde{\beta}||_2^2 \quad s.t. \ ||\tilde{\beta}||_1 \leq \tau \tag{4}$$

*for some $\tau > 0$. Therefore,*

$$\mathbb{E}[(\tilde{x}^T\beta - \tilde{x}^T\hat{\beta})^2] \lesssim \tau^2\sqrt{\frac{\log(B^2)}{n}} \tag{5}$$

*Proof.* This result is a straightforward generalization of Theorem 1 in Chatterjee [27] to the case of sub-Gaussian error terms. Hence, this proof is very similar. See Appendix E.1. □

Next we show that, under weak assumptions, the generalization risk of a Lasso-pruned forest is asymptotically no greater than that of its unpruned counterpart.

**Corollary 4.3.** *(Asymptotic advantage of Lasso pruning)*

*With the same (essentially assumption-free) setup as in Theorem 4.2, let $\beta_U = (1/B)(1, \ldots, 1)^T$ be the coefficient vector imposed by an unpruned forest. Let $\tilde{x} \in \mathbb{R}^B$ denote a vector containing the out-of-sample predictions by each of the trees in the given forest. Lastly, define for convenience $0 < c := \max(4\sqrt{2}\tau M\sigma, 16\sqrt{2}\tau^2 M^2) < \infty$. Then,*

$$\mathbb{E}[(\tilde{x}^T\beta - \tilde{x}^T\beta_U)^2] - \mathbb{E}[(\tilde{x}^T\beta - \tilde{x}^T\hat{\beta})^2] \geq \left(\mathbb{E}(\tilde{x})^T(\beta - \beta_U)\right)^2 - c\sqrt{\frac{\log(2B^2)}{n}}$$

*Proof.* By Theorem 4.2,

$$\mathbb{E}[(\tilde{x}^T\beta - \tilde{x}^T\hat{\beta})^2] \leq 2\tau M\sigma\sqrt{\frac{2\log(2B)}{n}} + 8\tau^2 M^2\sqrt{\frac{2\log(2B^2)}{n}} \leq c\sqrt{\frac{\log(2B^2)}{n}} \tag{6}$$

On the other hand, by Jensen's inequality,

$$\mathbb{E}[(\tilde{x}^T\beta - \tilde{x}^T\beta_U)^2] \geq \left(\mathbb{E}[\tilde{x}^T\beta - \tilde{x}^T\beta_U]\right)^2 = \left(\mathbb{E}(\tilde{x})^T(\beta - \beta_U)\right)^2 \tag{7}$$

Combining the above two equations concludes the proof. □

*Remark* 4.4. In the previous corollary we have proved, in passing, a simple lower bound for the generalization error of an unpruned random forest in terms of the discrepancy between the true regression coefficients $\beta$ and the uniform weights $\beta_U$ that an unpruned forest imposes. We see, then, that the success of Lasso pruning in large samples, is in general due to the fact that we learn the true weights of each tree in the forest, as opposed to keeping them uniform.

## 4.2. Generalization bound for Lasso-pruned regression forests

In this subsection and the next we will present a set of theoretical results that provide high-probability bounds for the out-of-sample risk of a pruned forest based on the observed in-sample risk and the complexity of the hypothesis space. To ease notation, we will use $X$ to denote the matrix of predictions, previously termed $\tilde{X} = F(X)$, bypassing the definition step each time an i.i.d. dataset $S = (X, Y)$ is observed.

The following theorem, which adapts Theorem 11.3 in Mohri et al. [28] to squared losses, provides a bound on the true risk of a hypothesis, denoted by $R(h)$, in terms of the empirical risk of said hypothesis ($\widehat{R}_S(h)$) and the Rademacher complexity of the corresponding hypothesis class.

**Theorem 4.5.** *Let $S = \{(x_1, y_1), \ldots, (x_n, y_n)\} \in (\mathcal{X} \times \mathcal{Y})^n$ be a dataset drawn i.i.d. from a distribution $\mathcal{D}$ over $\mathcal{X} \times \mathcal{Y}$. Assume squared losses are bounded by $M^2$, and hence $2M$-Lipschitz for any $y, y' \in \mathcal{Y}$ such that $|y - y'| \leq M$. Let $\mathcal{R}_n(\mathcal{H})$ and $\widehat{\mathcal{R}}_S(\mathcal{H})$ denote the Rademacher and empirical Rademacher complexities of a hypothesis class $\mathcal{H}$. Then, for all $h \in \mathcal{H}$ and any $\delta \in (0, 1)$, with probability at least $1 - \delta$*

$$R(h) \leq \widehat{R}_S(h) + M^2 \sqrt{\frac{\log(1/\delta)}{2n}} + 4M\mathcal{R}_n(\mathcal{H}) \tag{8}$$

*Proof.* The proof, which can be found in Appendix E.2, follows from a straightforward application of Talagrand's contraction lemma in combination with a classical Rademacher bound. $\square$

Combining Theorem 4.5 above with an adaptation of Theorem 11.15 in Mohri et al. [28] we obtain the main result in this subsection, a generalization bound for Lasso-pruned regression forests.

**Theorem 4.6.** *Let $\mathcal{X} \subseteq \mathbb{R}^B$ be the set of predictions of the label of an observation by a forest with $B$ trees. Let $\mathcal{Y} \subseteq \mathbb{R}$, $\mathcal{H}_{Lasso} = \{x \in \mathcal{X} \mapsto w \cdot x : ||w||_1 \leq \Lambda, w_j \geq 0 \forall j\}$, and $n$ is the sample size of the validation set, used to estimate the Lasso coefficients $w$. Define $r = \sup(|\inf(\mathcal{Y})|, \sup(\mathcal{Y}))$ and $M = \sup(\mathcal{Y}) - \inf(\mathcal{Y})$. Then, for all $h \in \mathcal{H}_{Lasso}$ and any $\delta \in (0, 1)$, with probability at least $1 - \delta$,*

$$R(h) \leq \widehat{R}_S(h) + M^2 \sqrt{\frac{\log(1/\delta)}{2n}} + 4r\Lambda M \sqrt{\frac{2\log(2B)}{n}} \tag{9}$$

*Proof.* See Appendix E.3. $\square$

*Remark* 4.7. In most practical cases, $r$ and $M$ are finite because the value of the target variable $y$ is bounded by physical constraints. Nonetheless, in practice even if $r$ or $M$ are infinite, finite proxies are possible and can be reasonable in many instances, as shown in our simulations. See Table 1 and 2 in Appendix F.

In Appendix D.0.1 we also provide similar generalization bounds for regression forests pruned by any combinatorics-based method, including SBS', SFS and BSF.

# 5. Testing conditions common to all experiments

As it is standard in the literature [12, 13, 29], we split our dataset into three parts: a training set (60%), a validation set (20%) and a testing set (20%). The first set is used to train the full decision forest, using $B$ bootstrap resamples (bagging). The second set is used to prune the full forest according to a particular method of choice. After sub-forest identification, the full forest is re-trained with the full 80% of the samples, corresponding to the training and validation sets. This ensures the comparison between the out-of-sample performances of the full forest and the alternative forest-pruning framework is unbiased. Lastly, the third set is used to compare the out-of-sample performance of the full forest and each of the pruning methods.

Statistical significance (at 5% significance level) of the difference in MSPE and number of trees used across different methods is assessed via Wilcoxon signed-rank tests [30], which is appropriate when normality cannot be assumed, and is the recommended choice in the literature on comparing predictor accuracies [17, 31].

In forest induction, random feature selection occurs just once at the start of tree induction: each tree has access only to a random subset of the features, itself with random cardinality (in expectation, 80% of the cardinality of the original feature set). This combination of bagging [32] with random subspaces [33] tends to outperform

both of them individually [34]. To fit the trees we use the most widespread decision tree model i.e. the standard CART algorithm [15] (`rpart` R library (version 4.1.19) [35]) with default parameters, which, at the time of writing, do not include pruning but they do include early stopping.

All tests are run on a 1.6 GHz Dual-Core Intel Core i5 processor with 8 GB of RAM. Lastly, a fixed random seed (123) is used across all simulations to maintain reproducibility and avoid unintentional bias.

# 6. Simulations with synthetic data

In this section, we test the four methods under consideration in 16 different scenarios, given by the following parameter combinations. Sample size: small (600), large (20,000); number of fully relevant variables: low (2), medium (8); number of trees in original forest: small (25), medium (100); noise level: low ($\sigma^2 = 0.04$), high ($\sigma^2 = 2$). We model the response, predictors and noise as follows: $Y = \sum_{j=1}^{10} \beta_j x_j + \varepsilon$, $\beta_j \in \{0, 1\}$, $\varepsilon \sim N(0, \sigma^2 I)$, $X := [x_1, x_2, \ldots, x_{10}] \sim N(0, I)$, i.e. assuming linearity (which may be a limitation of the proposed setup). In any case, observe that this setting is challenging for piecewise-constant trees, which present a strong inductive bias against additive models [36], making the success of forest pruning more remarkable.

The choice of $\sigma^2$ is motivated by the fact that, under the scenario with 2 variables and high noise, $\tilde{Y} := \sum_{j=1}^{10} \beta_j x_j \sim N(0, \tilde{\sigma}^2)$, where $\tilde{\sigma}^2 = \sum_{j=1}^{10} \beta_j^2 = 2$. Hence, $\tilde{Y} \sim N(0, 2) \perp \varepsilon \sim N(0, 2)$ and so $|\tilde{Y}|, |\varepsilon|$ are i.i.d., where independence follows from the fact that if two random variables are independent then any measurable function of them preserves independence. Hence, under such scenario, the probability that the noise will overpower the contribution from the regressors is $P(|\tilde{Y}| \leq |\varepsilon|) = 1/2$.

Each scenario (including the random data partitioning) is repeated 100 times to enhance reliability.

Our simulation results, which can be found in full detail in Appendix G, agree, qualitatively, with those in Zhang and Wang [12] and Bernard et al. [13], namely, that it is possible to considerably downsize a given forest while retaining or even improving (sometimes, substantially) out-of-sample performance.

Next we show the out-of-sample MSPE of unpruned and pruned forests in the worst (small sample, high noise) and best (large sample, low noise) scenarios. See Appendix G.1 for the remaining results.

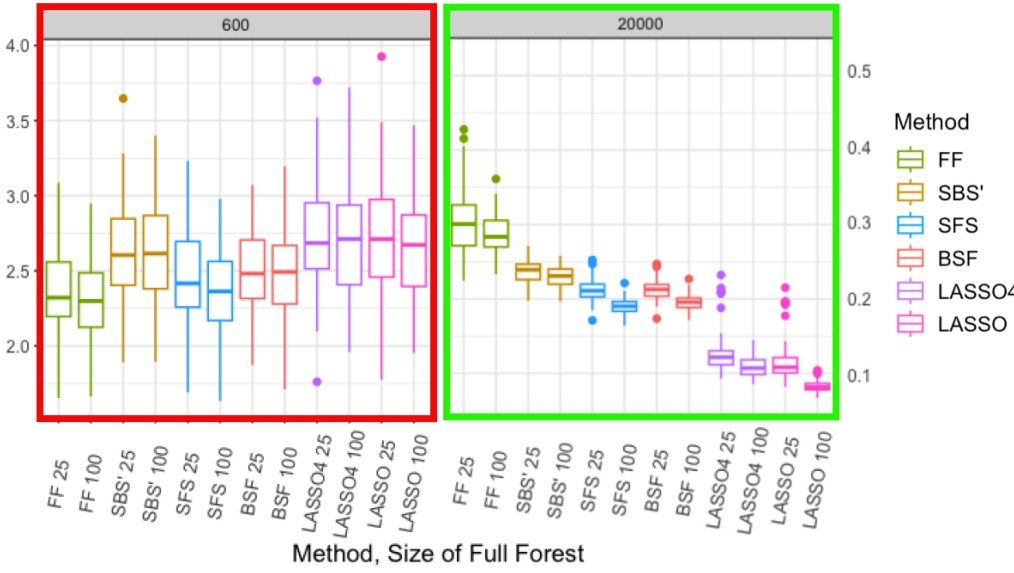

Figure 2: MSPE of full forest (FF) and forest-pruning methods under the worst (left) and best (right) conditions

In all of the low-noise scenarios (8 out of 8), there is at least one method tested that yields a statistically significant out-of-sample reduction in MSPE, the largest reduction being as large as $71.4\%$, by Lasso using 1/5 of the full forest trees under scenario 6 ($n = 20000$, 2 variables, $B = 100$, $\sigma_\varepsilon^2 = 0.04$).

The performance improvement is less impressive under the high-noise regime ($\sigma_\varepsilon^2 = 2$, 50 times higher than the low-noise setup), but we still observed statistically significant out-of-sample MSPE reductions in 5 out of 8 scenarios, the largest reduction being $36\%$, by Lasso using 2/5 of the full forest trees under scenario 16 ($n = 20000$, 8 variables, $B = 100$, $\sigma_\varepsilon^2 = 2$). Note: 'LASSO4' restricts Lasso to selecting at most 4 trees.

# 7. Experiments with real data

In this section, we test the four forest-pruning methods under consideration on three different real datasets of diverse sizes, complexities and feature types: the well-known bult-in Iris dataset in R (150 samples, 4 features, 1 response), as well as the Diamonds dataset (53,940 samples, 9 features, 1 response), and the Midwest dataset (437 samples, 27 features, 1 response), both available in the `ggplot2` library (version 3.4.3) [37]. We repeat the entire testing procedure 100 times for each of the three datasets.

As we have seen, larger sample sizes tend to favor forest pruning. Hence, one may anticipate that forest pruning might suffer in datasets with only a modest number of samples (e.g. the Iris and Midwest datasets). We decided to test this hypothesis and show that even under small sample settings forest pruning can still be relatively successful. Results for the Iris and Midwest datasets can be found in Appendix G.3.

The Diamonds dataset contains the prices and other 9 attributes (6 continuous and 3 ordinal) of nearly 54,000 diamonds. We predict diamond price as a function of carats, cut quality, color and clarity, among other variables. Given the size of this dataset, we only consider a random $40\%$ of the nearly 54,000 observations (21,576). We use $B = 200$ for all methods. Appendix H contains more details about this dataset.

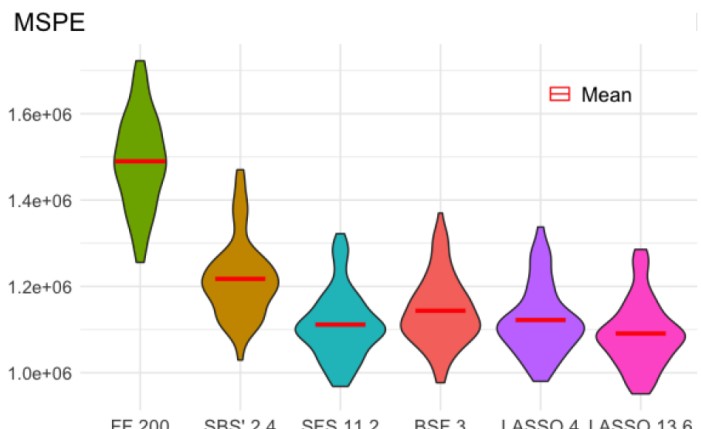

Figure 3: MSPE and average # trees (across 100 repetitions) in full and pruned forests in Diamonds

The results obtained, fueled by a large sample size, are remarkable, with reductions in out-of-sample MSPE ranging roughly from $18\%$ to $27\%$, while using just a fraction of the trees in the original forest. Due to the favorable conditions of this dataset, we can obtain excellent results by selecting just 3 trees, as the BSF results show (Table 37), which this time was constrained to search through (at most) all 3-tree combinations in the decision forest, instead of the usual 4. Indeed, BSF reduced out-of-sample MSPE by $23.3\%$ while using just 3 trees instead of 200 (-98.5\%). However, it was the slowest among the methods surveyed (7 minutes), while the Lasso-based methods just took around one second.

# 8. From sub-forest to tree

Pruning a decision forest of a few hundred trees down to, say, a few tens, is a step in the right direction towards making decision forest interpretable, as it dramatically reduces the number of feature splits (logic rules) that would occur in a tree that merges the selected trees. In addition, if the number of selected trees is small enough, such a merged tree could be interpretable, as opposed to the initial, uninterpretable decision forest. Even though reducing the entire ensemble to just one (merged) base learner of manageable complexity is essential to obtaining interpretability, to our knowledge this has not been pursued in the ensemble-pruning literature

before. We conjecture the reason for this is that the emphasis has traditionally been skewed towards accuracy rather than interpretability.

Note that there exists a one-to-one correspondence between the weighted sum of the predictions of each tree in a collection and those of a single, deeper tree, allowing us to merge several trees into one this way. Thus, as long as the resulting tree is not too large, we will have an interpretable tree that retains all the advantages of the sub-forest. In Appendix J we show our process to combine the selected trees into one, as in Quinlan [38].

The out-of-sample performance of a pruned tree will tend to be worse than that of a pruned forest, as the pruned forest, even if it may look like a tree, is in fact a (small) forest. On the other hand, it is true that the pruned forest will usually be more complex than the pruned tree, as it combines a few trees from the original forest (which is precisely what gives it the accuracy advantage over the pruned tree). However, if (local) complexity is kept below a certain subjective threshold then only the gain in accuracy is of consequence.

As an illustration, we revisit the Diamonds dataset. On this occasion, we compare the first tree in the forest induction process, fitted via CART (pruned by way of early stopping), which had access to all features, versus the reduced ensemble obtained by BSF (merging two trees, each having access only to a random subset of the features). Figure 4 shows that the out-of-sample accuracy of a BSF-pruned forest of just 2 trees tends to be substantially higher than that of a traditional pruned tree (-34.2% in MSPE on average, p-value $< 10^{-16}$) as well as that of the starting full forest of 200 trees.

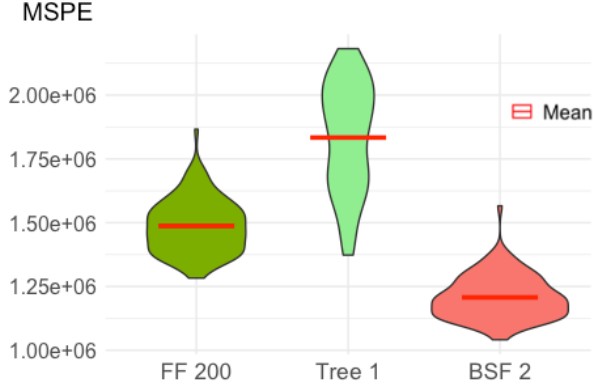

Figure 4: MSPE of the full forest, pruned tree and BSF-pruned forest in the Diamonds dataset

As shown in Appendix J, even if globally the pruned forest is clearly more complex than the pruned tree, locally the increase in complexity can be minimal – and in all cases many orders of magnitude simpler than the full forest (which remains a black box), while being more accurate on average.

## 9. Conclusion

In this work we have revisited forest pruning in the context of regression, and have shed new light onto when and why it works from a theoretical perspective: we have proved the asymptotic superiority of Lasso-pruned forests over their unpruned counterparts, as well as high-probability finite-sample generalization bounds for the main forest pruning methods, including a simple (yet often powerful) one that we have introduced (BSF).

In addition, we have contributed significant empirical evidence (16 synthetic scenarios and 3 real datasets) to the current body of empirical results in the forest-pruning literature. In the vast majority of cases considered (16 out of 19), there is at least one forest-pruning method that, in expectation, provides a net performance gain compared to the decision forest: equivalent or significantly better out-of-sample MSPE while using a small fraction of the trees – in some cases, so small that the trees can be meaningfully combined into a single tree for maximum interpretability.

Forest pruning shines under large sample sizes and high signal-to-noise ratios, with Lasso pruning being particularly effective in such scenario. However, as in the case of variable selection [39], no forest-pruning method dominates under all circumstances. Thus, cross-validation, though potentially expensive, is advised.

# Acknowledgements

I would like to express sincere gratitude to Wei-Yin Loh and Kris Sankaran for their insightful suggestions, which significantly improved the quality of this paper. Likewise, I thank the anonymous reviewers for their comments and constructive criticism. All remaining errors are solely my own.

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

# A. Pseudo-code for forest-pruning algorithms

---

**Algorithm 1** Sequential Forward Selection (SFS)

---

1: **for** $i = 1 : B$ **do**
2:     **for** $j = 1 : (B + 1 - i)$ **do**
3:         Compute MSPE of sub-forest after adding tree $j$ to current subset
4:     **end for**
5:     Find added tree that yielded minimum MSPE and update sub-forest
6:     Update full forest (remove the tree added to sub-forest)
7: **end for**
8: **return** Added tree indices at minimum MSPE across all iterations

---

---

**Algorithm 2** Modified Sequential Backward Selection (SBS')

---

1: Compute MSPE of full forest
2: **for** $i = 1 : B$ **do**
3:     **for** $j = 1 : (B + 1 - i)$ **do**
4:         Compute MSPE of forest after deleting tree $j$
5:     **end for**
6:     Compute abs. MSPE difference of current forest and each MSPE computed in inner loop
7:     Find deleted tree that yielded minimum difference
8:     Update sub-forest by deleting least relevant tree and update sub-forest MSPE
9: **end for**
10: **return** Surviving tree indices at minimum MSPE across all iterations

---

---

**Algorithm 3** Best Sub-Forest (BSF)

---

1: **for** $k = 1 : K$ **do**
2:     $N = \binom{B}{k}$
3:     **for** $j = 1 : N$ **do**
4:         Compute MSPE of $j$-th combination of $k$ trees.
5:         **if** MSPE < current min MSPE **then**
6:             Update min MSPE
7:         **end if**
8:     **end for**
9: **end for**
10: **return** Optimal tree indices

---

---

**Algorithm 4** Lasso pruning

---

1: Fit Lasso coefficients with non-negativity constraints and no intercept
2: Perform 10-fold cross-validation to select $\lambda$
3: **if** # nonzero coefficients > desired threshold **then**
4:     Sort fitted coefficients in descending order
5:     Set upper bound to 0 for all tree indices exceeding threshold
6:     Repeat steps 1 and 2 with new zero upper bounds
7: **end if**
8: **return** Optimal coefficients

---

## B. Ghost tree selection

This method is inspired by the Ghost variable method laid out in Delicado and Peña [18], which is a state-of-the-art model-agnostic method used for conditional variable importance scoring after model deployment.

The Ghost variable method measures the relevance of each variable in a model conditional on the presence of the rest of variables by comparing the predictions of the original model in the test set with those obtained when the variable of interest is substituted in the test set by its 'Ghost' representation , defined as the prediction of this variable by using the rest of explanatory variables. When a linear model is used to obtain the ghost representation, the mechanism is the same as that seen in the calculation of the variance inflation factor of a variable in the context of multiple linear regression.

After further analysis, we realized that, in general (not just in a forest pruning context), the model used to obtain the Ghost representation must not always be a linear model, but instead a model whose capacity is commensurate with the capacity of the predictive model at hand to combine its predictors.

Next show that, with a correctly capacity-calibrated model to obtain the Ghost representation of a given tree in the forest, the Ghost Tree method reduces to Sequential Backward Selection: if below we substitute $\hat{t}_j$ by $\frac{1}{B-1}\sum_{i \neq j}\hat{t}_i$, the new forest prediction is

$$\frac{1}{B}\sum_{j=1}^{B}\hat{t}_j = \frac{1}{B}\left(\frac{1}{B-1}\sum_{i \neq j}\hat{t}_i + \sum_{i \neq j}\hat{t}_i\right) = \frac{1}{B}\left(\frac{B}{B-1}\sum_{i \neq j}\hat{t}_i\right) = \frac{1}{B-1}\sum_{i \neq j}\hat{t}_i \tag{10}$$

which, under a 'Ghost' importance metric involving the MSPE, coincides exactly with Sequential Backward Selection, where one measures the increase in MSPE after excluding the $j$-th tree.

## C. Background definitions

Below is a list of definitions that the reader should be familiar with.

**Definition C.1.** (Bounded problem) Let $\mathcal{Y} \subseteq \mathbb{R}$ be a label space. A regression problem is said to be bounded if the loss function $\ell : \mathcal{Y} \times \mathcal{Y} \to \mathbb{R}_+$ involved is bounded, i.e. $\exists M > 0$ such that $\ell(y, y') \leq M$ for all $y, y' \in \mathcal{Y}$.

**Definition C.2.** (Generalization risk) Given a feature space $\mathcal{X}$, a label space $\mathcal{Y}$, a hypothesis $h \in \mathcal{H}$, a loss function $\ell : \mathcal{Y} \times \mathcal{Y} \to \mathbb{R}_+$ and an underlying distribution $\mathcal{D}$ on $(\mathcal{X}, \mathcal{Y})$, the generalization risk (or error) of $h$ is defined as

$$R(h) := \mathbb{E}_{(x,y)\sim\mathcal{D}}[\ell(h(x), y)] \tag{11}$$

**Definition C.3.** (Empirical risk) Given a feature space $\mathcal{X}$, a label space $\mathcal{Y}$, a hypothesis $h \in \mathcal{H}$, a loss function $\ell : \mathcal{Y} \times \mathcal{Y} \to \mathbb{R}_+$ and a sample $S = \{(x_1, y_1), \ldots, (x_n, y_n)\}$, the empirical risk (or error) of $h$ is defined as the method of moments approximation of the generalization risk, i.e.

$$\widehat{R}(h) := \frac{1}{n}\sum_{i=1}^{n}\ell(h(x_i), y_i) \tag{12}$$

**Definition C.4.** (PAC-learnable) A function class is said to be PAC-learnable if there exists an algorithm $\mathcal{A}$ and a polynomial map $poly()$ such that for any $\varepsilon > 0$ and $\delta > 0$, for all distributions $\mathcal{D}$ on $(\mathcal{X}, \mathcal{Y})$ and for any target function $f \in \mathcal{F}$ such that $f : \mathcal{X} \to \mathcal{Y}$, the following holds for any sample size $n \geq poly(1/\varepsilon, 1/\delta, size(x), size(f))$:

$$\mathbb{P}_{S\sim\mathcal{D}^n}(R(h_S) \leq \varepsilon) \geq 1 - \delta \tag{13}$$

where $size(f)$ and $size(x)$ denote, respectively, the maximal cost of the computational representation of $f \in \mathcal{F}$ and any $x \in \mathcal{X}$.

The above definition, which is a slightly simplified adaptation of Definition 2.3 in Mohri et al. [28], loosely speaking refers to the capability of a function class to allow a certain algorithm, after observing a large enough sample of points, to return a hypothesis that, with high Probability (at least $1 - \delta$), is Approximately Correct (error at most $\varepsilon$). Such an algorithm is then called a PAC-learning algorithm for the function class $\mathcal{F}$.

**Definition C.5.** (Rademacher distribution) The Rademacher distribution, named after German-American mathematician Hans Rademacher, is a discrete distribution with the following probability mass function:

$$f(x) = \begin{cases} 1/2, & \text{if } x = -1 \\ 1/2, & \text{if } x = 1 \\ 0, & \text{otherwise} \end{cases} \tag{14}$$

**Definition C.6.** (Empirical Rademacher complexity) Let $S = \{z_1, \ldots, z_n\} \subseteq \mathcal{Z}^n$ be a fixed sample of size $n$ and consider a function class $\mathcal{G}$ of real-valued functions over $\mathcal{Z}$. Let $\sigma = (\sigma_1, \ldots, \sigma_n)^T$ be an $n$-dimensional vector of independent Rademacher random variables $\sigma_i$. Then, the empirical Rademacher complexity of $\mathcal{G}$ with respect to the sample $S$ is defined as

$$\widehat{\mathcal{R}}_S(\mathcal{G}) := \frac{1}{n} \mathbb{E}_\sigma \left[ \sup_{g \in \mathcal{G}} \sum_{i=1}^n \sigma_i g(z_i) \right] \tag{15}$$

Recall that for any two vectors $a, b$ with angle $\theta$ between them, $a^T b = \cos(\theta) ||a||_2 ||b||_2$; if those vectors have mean zero, their cosine coincides with their correlation coefficient. Therefore, intuitively, the empirical Rademacher complexity measures how well the elements in the function class $\mathcal{G}$ could correlate with random Radamacher vectors. Such random vectors can be viewed as random arrangements of labels in a classification context.[1] Richer (i.e. more *complex*) function classes will be able to correlate more with random vectors.

**Definition C.7.** (Rademacher complexity) Let $\mathcal{D}$ denote the distribution according to which samples are drawn. For any integer $n \geq 1$, the Rademacher complexity of $\mathcal{G}$ is defined as the expectation of the empirical Rademacher complexity over all samples of size $n$ drawn according to $\mathcal{D}$, i.e.

$$\mathcal{R}_n(\mathcal{G}) := \mathbb{E}_{S \sim \mathcal{D}^n} \left[ \widehat{\mathcal{R}}_S(\mathcal{G}) \right] \tag{16}$$

**Definition C.8.** ($\sigma$-sub Gaussian random variable) A random variable $X$ is called $\sigma$-sub Gaussian if it satisfies $\mathbb{E}\left[ e^{t(X - \mathbb{E}X)} \right] \leq e^{\frac{t^2 \sigma^2}{2}}$ for all $t \in \mathbb{R}$. Equivalently, $X$ is $\sigma$-sub Gaussian if $\exists \sigma > 0$ such that $\mathbb{P}(|X| > t) \leq 2e^{-\frac{t^2}{2\sigma^2}}$ for all $t > 0$.

# D. Further theoretical results

**Lemma D.1.** *The size of the estimated $\hat{\beta}$ coefficients in Lasso and, in particular, non-negative Lasso, is not always a monotonic function of the penalty parameter $\lambda \geq 0$.*

*Proof.* We can prove the result both for Lasso and non-negative Lasso by focusing on the case where the solution lies in the interior of the feasible region defined by the non-negativity constraints on the $\beta$ parameters, which then become redundant for any value of $\lambda$. This is the case, for example, when $y$ is a positive function of a set of (possibly) correlated regressors with large enough signal-to-noise ratio: for any $j$, reducing $\beta_j \geq 0$ beyond zero is not optimal for any value of $\lambda$, as it increases both the residual sum of squares (as it would lead to a deviation from the observed response vector $y$) and the penalty term (as it would deviate from 0). Therefore, it suffices to consider this particular case to prove the lemma.

Hence, we wish to solve the following unconstrained convex quadratic program

$$\min_{\beta \in \mathbb{R}_+^p} ||y - X\beta||_2^2 + \lambda \beta^T e \tag{17}$$

where $e := \{1\}^p$. Then, we can find the unique solution by considering first order conditions:

Let $\mathcal{L} := ||y - X\beta||_2^2 + \lambda \beta^T e$ then

$$\nabla \mathcal{L}(\beta) = -2X^T y + 2X^T X\beta + \lambda e \overset{\text{set}}{=} 0 \Rightarrow X^T X\hat{\beta} = X^T y - \frac{\lambda}{2} e \tag{18}$$

---

[1] Although the notion of Rademacher complexity exists more generally – in particular, it can be applied as well in regression settings, where it might be more intuitive to consider e.g. random Gaussian vectors, but Rademacher random variables, being bounded with modulus 1, are more convenient to use in derivations.

For simplicity, suppose in our example $X^T X$ is invertible (which usually is, as it's related to the regressor sample covariance matrix, positive semi-definite by construction, and positive definite unless degenerate), then

$$\hat{\beta} = (X^T X)^{-1} X^T y - \frac{\lambda}{2} (X^T X)^{-1} e$$

$$= \hat{\beta}^{OLS} - \frac{\lambda}{2} (X^T X)^{-1} e$$

where $\hat{\beta}^{OLS}$ denotes the estimated regression coefficients through ordinary least squares.

Hence,

$$\frac{\partial \hat{\beta}}{\partial \lambda} = -\frac{1}{2} (X^T X)^{-1} e \tag{19}$$

Therefore, whether $\hat{\beta}_j$ for any given $j \in \{1, 2, \ldots, p\}$ increases or decreases as $\lambda$ increases depends on the correlation structure of the regressors. In particular, if $X^T X = I$ (orthonormal regressors), then we recover a well-known special case that has the closed-form Lasso solution $\hat{\beta}_j = \text{sign}(\hat{\beta}_j^{OLS})(|\hat{\beta}_j^{OLS}| - \tilde{\lambda})^+$ (Tibshirani, 1996), with $\tilde{\lambda} := \lambda/2$, from which it can be deduced that for any $j \in \{1, 2, \ldots, p\}$, $\hat{\beta}_j$ decreases monotonically as $\lambda$ increases, as we can also see in our derivation under that orthonormality condition. $\square$

Next we state the following Rademacher complexity bound for linear hypotheses with bounded $\ell_1$ norm, adapted from Mohri et al. [28] (Theorem 11.15).

**Theorem D.2.** *Let $\mathcal{X} \subseteq \mathbb{R}^p$, $\mathcal{Y} \subseteq \mathbb{R}$ and $S = \{(x_1, y_1), \ldots, (x_n, y_n)\} \in (\mathcal{X} \times \mathcal{Y})^n$ be a dataset drawn i.i.d. from a distribution $\mathcal{D}$ over $\mathcal{X} \times \mathcal{Y}$. Assume $\exists r > 0$ such that, for all $i \in [n]$, $||x_i||_\infty \leq r$, and let $\mathcal{H} = \{x \in \mathcal{H} \mapsto w \cdot x : ||w||_1 \leq \Lambda\}$. Then, the empirical Rademacher complexity of $\mathcal{H}$ satisfies*

$$\widehat{\mathcal{R}}_S(\mathcal{H}) \leq \sqrt{\frac{2r^2 \Lambda^2 \log(2p)}{n}} \tag{20}$$

*Proof.* For all $i \in [n] \equiv \{1, 2, \ldots, n\}$, denote the $j$-th component of the vector $x_i$ by $x_{i,j}$. Then,

$$\widehat{\mathcal{R}}_S(\mathcal{H}) = \frac{1}{n} \mathbb{E}_\sigma \left[ \sup_{w: ||w||_1 \leq \Lambda} \sum_{i=1}^n \sigma_i w \cdot x_i \right]$$

$$= \frac{\Lambda}{n} \mathbb{E}_\sigma \left[ \left\| \sum_{i=1}^n \sigma_i x_i \right\|_\infty \right] \quad \text{by definition of dual norm}$$

$$= \frac{\Lambda}{n} \mathbb{E}_\sigma \left[ \max_{j \in [p]} \left| \sum_{i=1}^n \sigma_i x_{i,j} \right| \right] \quad \text{by definition of } ||\cdot||_\infty$$

$$= \frac{\Lambda}{n} \mathbb{E}_\sigma \left[ \max_{j \in [p]} \max_{s \in \{-1, 1\}} s \sum_{i=1}^n \sigma_i x_{i,j} \right]$$

$$= \frac{\Lambda}{n} \mathbb{E}_\sigma \left[ \sup_{z \in \mathcal{A}} \sum_{i=1}^n \sigma_i z_i \right]$$

Where $\mathcal{A} = \{s[x_{1,j}, x_{2,j}, \ldots, x_{n,j}]^T : j \in [p], s \in \{-1, 1\}\}$. Note $\forall z \in \mathcal{A}$, $||z||_2 \leq \sqrt{nr^2} = r\sqrt{n}$. Thus, by Massart's lemma, since $\mathcal{A}$ contains at most $2p$ elements,

$$\widehat{\mathcal{R}}_S(\mathcal{H}) \leq r\Lambda \sqrt{\frac{2 \log(2p)}{n}} \tag{21}$$

$\square$

### D.0.1. Generalization bounds for regression forests pruned with combinatorics

Similar to the main result in the previous subsection about Lasso-pruned regression forests, next we provide generalization bounds for regression forests pruned by any combinatorics-based method, including SBS', SFS and BSF. Note that in this case we are dealing with a finite hypothesis class, as there is only a finite number of possibilities the algorithm considers: unlike in Lasso, in this case the weights are predefined to be uniform over the selected trees.

**Theorem D.3.** *(Generalization bound for finite hypothesis classes) Let $\ell$ be a loss function and suppose there exists $M > 0$ such that $\ell(h(x), y) \leq M$ for all $h \in \mathcal{H}$ and $(x, y) \in \mathcal{X} \times \mathcal{Y}$, where $|\mathcal{H}| < \infty$. Then, for any $\delta \in (0, 1)$, with probability at least $1 - \delta$,*

$$R(h) \leq \widehat{R}_S(h) + M\sqrt{\frac{|\mathcal{H}| + \log(1/\delta)}{2n}} \tag{22}$$

*Proof.* See Theorem 11.1 in Mohri et al. [28]. $\qquad\square$

Focusing now on the specific context of forest pruning, we have the following result for BSF.

**Theorem D.4.** *Let $\mathcal{X} \subseteq \mathbb{R}^B$ be the set of predictions of the label of an observation by a forest with $B$ trees. Let $\mathcal{Y} \subseteq \mathbb{R}$, $\mathcal{H}_{BSF} = \{x \in \mathcal{X} \mapsto we \cdot x : w \in \{1, 1/2, \dots, 1/K\}, e \in \{0, 1\}^B, \sum_{i=1}^{B} e_i = w^{-1}\}$, and $n$ is the sample size of the validation set. Assume $K \leq B/2$. Define $M = \sup(\mathcal{Y}) - \inf(\mathcal{Y})$. Then, for all $h \in \mathcal{H}_{BSF}$ and any $\delta > 0$, with probability at least $1 - \delta$,*

$$R(h) \leq \widehat{R}_S(h) + M^2\sqrt{\frac{K \log B + \log \frac{1}{\delta(K-1)!}}{2n}} \tag{23}$$

*Proof.* The proof relies on Hoeffding's inequality, the union bound and straightforward combinatorics. See Appendix E.4. $\qquad\square$

*Remark* D.5. It can be proved (see Appendix D.7) that BSF pruning is $K$-optimal within the class of combinatorics-based tree selection methods.

A similar bound can be derived for forests pruned with SBS' or SFS.

**Theorem D.6.** *Let $\mathcal{X} \subseteq \mathbb{R}^B$, with $B$ even, be the set of predictions of the label of an observation by a forest with $B$ trees. Let $\mathcal{Y} \subseteq \mathbb{R}$, $\mathcal{H} = \{x \in \mathcal{X} \mapsto we \cdot x : w \in \{1, 1/2, \dots, 1/B\}, e \in \{0, 1\}^B, \sum_{i=1}^{B} e_i = w^{-1}\}$ and $n$ is the sample size of the validation set. Define $M = \sup(\mathcal{Y}) - \inf(\mathcal{Y})$. Then, for all $h \in \mathcal{H}$ and any $\delta > 0$, with probability at least $1 - \delta$,*

$$R(h) \leq \widehat{R}_S(h) + M^2\sqrt{\frac{\frac{B}{2} \log B + \log \frac{1}{\delta(B/2-1)!} + \log(2)}{2n}} \tag{24}$$

*Proof.* The proof is very similar to that of Theorem D.4 with a correspondingly different hypothesis space, and can be found in Appendix E.4.1. $\qquad\square$

In Table 1 and 2 in the Appendix, we show the results of our simulations, which validate the theoretical generalization bounds derived. The scenarios tested are similar to those discussed in Section 6, i.e. a response that depends on several i.i.d. Gaussian covariates and Gaussian noise. On this occasion, the train-validation-test split is 0.3:0.35:0.35, as we try to empirically analyze the effect of the forest size $B$ and sample size $n$ on the generalization bounds, shifting focus to the validation and test sets.

**Lemma D.7.** *Let $\delta \in (0, 1)$. With probability at least $1 - \delta$, BSF pruning is $K$-optimal out of sample within the class of combinatorics-based tree selection methods.*

*Proof.* For a fixed sub-forest size, the empirical risk $\widehat{R}_s(\hat{h})$ for the BSF empirical risk minimizer $\hat{h}$ is by construction guaranteed to be no greater (usually smaller) than that of any other combinatorics-based tree selection method, having hypothesis class $\mathcal{H}$. Moreover, for $K \leq B$, $|\mathcal{H}_{BSF}| = \sum_{j=1}^{K} \binom{B}{j} \leq \sum_{j=1}^{B} \binom{B}{j} = |\mathcal{H}|$. By Theorem D.3, combining both facts concludes the proof. $\qquad\square$

# E. Complete proofs to theoretical results

## E.1. Proof of Theorem 4.2

Notation: let $\tilde{x} = F(x) \in \mathbb{R}^B$ be an out-of-sample vector of forest predictions, independent (and with identical distribution) from the in-sample data $Y \in \mathbb{R}^n$, $\tilde{X} = (\tilde{X}_1, \ldots, \tilde{X}_B) \in \mathbb{R}^{n \times B}$. Then, define $V_{j,k} = \mathbb{E}(\tilde{X}_j \tilde{X}_k) - \frac{1}{n} \sum_{i=1}^n \tilde{X}_{i,j} \tilde{X}_{i,k}$. Lastly, let $\mathcal{F}$ be the sigma field generated by $(\tilde{X}_{i,j})$ for $1 \leq i \leq n$ and $1 \leq j \leq B$.

First we bound $\mathbb{E}\left[ ||\tilde{X}\beta - \tilde{X}\hat{\beta}||_2^2 \right]$ as follows. By convexity of the feasible region, $(\tilde{X}\beta - \tilde{X}\hat{\beta})^T (Y - \tilde{X}\hat{\beta}) \leq 0$, which can be written as

$$||\tilde{X}\beta - \tilde{X}\hat{\beta}||_2^2 \leq \sum_{j=1}^B (\hat{\beta}_j - \beta_j) \left( \sum_{i=1}^n \varepsilon_i \tilde{X}_{i,j} \right) \tag{25}$$

$$\leq 2\tau ||\tilde{X}^T \varepsilon||_\infty \tag{26}$$

Since $|\tilde{X}_{i,j}| \leq M$ a.s. for all $i, j$, and by the definition of sub-Gaussian random variable, $\mathbb{E}(||\tilde{X}^T \varepsilon||_\infty) \leq M\sigma\sqrt{2n \log(2B)}$, which implies

$$\mathbb{E}\left[ ||\tilde{X}\beta - \tilde{X}\hat{\beta}||_2^2 \right] \leq 2\tau M\sigma\sqrt{2n \log(2B)} \tag{27}$$

Now, observe that $|\tilde{X}_{i,j}| \leq M$ a.s. for all $i, j$ implies $|\mathbb{E}(\tilde{X}_j \tilde{X}_k) - \tilde{X}_{i,j} \tilde{X}_{i,k}| \leq 2M^2$ for all $i, j, k$. Then, by Hoeffding's inequality, for any $\alpha \in \mathbb{R}$, $\mathbb{E}(e^{\alpha V_{j,k}}) \leq e^{2\alpha^2 M^4/n}$. Hence,

$$\mathbb{E}\left( \max_{1 \leq j,k \leq B} |V_{j,k}| \right) \leq 2M^2 \sqrt{\frac{2\log(2B^2)}{n}} \tag{28}$$

Then,

$$\mathbb{E}\left[ \mathbb{E}\left( (\tilde{x}^T \beta - \tilde{x}^T \hat{\beta})^2 - \frac{1}{n}||\tilde{X}\beta - \tilde{X}\hat{\beta}||_2^2 \mid \mathcal{F} \right) \right] = \mathbb{E}\left[ \sum_{j=1}^B \sum_{k=1}^B (\beta_j - \hat{\beta}_j)(\beta_k - \hat{\beta}_k) V_{j,k} \right] \tag{29}$$

$$\leq \mathbb{E}\left[ 4\tau^2 \max_{1 \leq j,k \leq B} |V_{j,k}| \right] \tag{30}$$

$$\leq 8\tau^2 M^2 \sqrt{\frac{2\log(2B^2)}{n}} \tag{31}$$

Combining the above with inequality 27 above concludes the proof.

$\square$

## E.2. Proof of Theorem 4.5

*Proof.* Our proof structure closely follows the one for Theorem 11.3 in Mohri et al. [28].

Let $\sigma_i, \forall i \in \{1, 2, \ldots, n\}$ be i.i.d. Rademacher random variables, and let $\mathcal{L} = \{(x, y) \mapsto \ell(h(x), y) : h \in \mathcal{H}\}$. Since, by assumption, for any fixed $y_i$, $y \mapsto \ell(y, y_i)$ is $2M$-Lipschitz, by Talagrand's contraction lemma, we can write:

$$\widehat{\mathcal{R}}_S(\mathcal{L}) = \frac{1}{n}\mathbb{E}_\sigma\left[ \sum_{i=1}^n \sigma_i \ell(h(x_i), y_i) \right] \leq \frac{1}{n}\mathbb{E}_\sigma\left[ \sum_{i=1}^n \sigma_i 2M h(x_i) \right] = 2M\widehat{\mathcal{R}}_S(\mathcal{H}) \tag{32}$$

Now, consider the following Rademacher bound, which is a straightforward adaptation of Theorem 3.3 in Mohri et al. [28], where the unit upper bound has been replaced by $M^2$.

**Theorem E.1.** *Let $\mathcal{L}$ be a family of functions mapping from a set $\mathcal{Z}$ to $[0, M^2]$. Then, for any $\delta \in (0, 1)$, with probability at least $1 - \delta$ over the draw of an i.i.d. sample $S$ of size $n$, the following holds for all $\ell \in \mathcal{L}$:*

$$E[\ell(z)] \leq \frac{1}{n} \sum_{i=1}^n \ell(z_i) + M^2 \sqrt{\frac{\log(1/\delta)}{2n}} + 2\mathcal{R}_n(\mathcal{L}) \tag{33}$$

*and,*

$$E[\ell(z)] \leq \frac{1}{n}\sum_{i=1}^{n}\ell(z_i) + 3M^2\sqrt{\frac{\log(2/\delta)}{2n}} + 2\widehat{\mathcal{R}}_S(\mathcal{L}) \tag{34}$$

*Proof.* The proof mimics that of Theorem 3.3 in Mohri et al. [28] replacing the codomain's unit upper bound by $M^2$. $\qquad\square$

Therefore, using the standard definitions of true and empirical risk and applying the bound in equation 32 to Theorem E.1 above we get the desired result. $\qquad\square$

## E.3. Proof of Theorem 4.6

*Proof.* By part 1 of Theorem 4.5, for all $h \in \mathcal{H}_{Lasso}$ and any $\delta \in (0,1)$, with probability at least $1 - \delta$,

$$R(h) \leq \widehat{R}_S(h) + M^2\sqrt{\frac{\log(1/\delta)}{2n}} + 4M\mathcal{R}_n(\mathcal{H}_{Lasso})$$

By Theorem D.2 with $p = B$,

$$\widehat{\mathcal{R}}_S(\mathcal{H}_{Lasso}) \leq r\Lambda\sqrt{\frac{2\log(2B)}{n}} \tag{35}$$

which implies

$$\mathcal{R}_n(\mathcal{H}_{Lasso}) = \mathbb{E}[\widehat{\mathcal{R}}_S(\mathcal{H}_{Lasso})] \leq r\Lambda\sqrt{\frac{2\log(2B)}{n}} \tag{36}$$

Therefore, we have

$$R(h) \leq \widehat{R}_S(h) + M^2\sqrt{\frac{\log(1/\delta)}{2n}} + 4r\Lambda M\sqrt{\frac{2\log(2B)}{n}} \tag{37}$$

$\qquad\square$

## E.4. Proof of Theorem D.4

Because $K \leq B/2$,

$$|\mathcal{H}_{BSF}| = \sum_{j=1}^{K}\binom{B}{j} \leq K\binom{B}{K} = \frac{1}{(K-1)!}B(B-1)\cdots(B-(K-1)) \leq \frac{B^K}{(K-1)!} \tag{38}$$

Since $K$ is typically small (e.g. 2 to 4), the price of the above inequalities in return for a much more insightful and practical bound seems adequate. Therefore, $\log(|\mathcal{H}_{BSF}|) \leq K\log(B) - \log((K-1)!)$

Now, let $S := \sum_{i=1}^{n}(h(x_i) - y_i)^2$, where by assumption, for all $i \in \{1,\ldots,n\}$ and any $h \in \mathcal{H}_{BSF}$, $(h(x_i) - y_i)^2 \leq M^2$ with probability one. Then, by Hoeffding's inequality, for any $\varepsilon > 0$ and any $h \in \mathcal{H}_{BSF}$,

$$\mathbb{P}(\mathbb{E}(S) - S > n\varepsilon) = \mathbb{P}(R(h) - \widehat{R}_S(h) > \varepsilon) \leq \exp\left(\frac{-2n\varepsilon^2}{M^4}\right) \tag{39}$$

Then, by the union bound, we have

$$\mathbb{P}(\exists h \in \mathcal{H}_{BSF} : R(h) - \widehat{R}_S(h) > \varepsilon) \leq \sum_{h \in \mathcal{H}_{BSF}}\mathbb{P}(R(h) - \widehat{R}_S(h) > \varepsilon) \leq \frac{B^K}{(K-1)!}\exp\left(\frac{-2n\varepsilon^2}{M^4}\right) \tag{40}$$

Set $\frac{B^K}{(K-1)!}\exp\left(\frac{-2n\varepsilon^2}{M^4}\right) = \delta$ and solve for $\varepsilon$ to get

$$\varepsilon = M^2\sqrt{\frac{K\log B + \log\frac{1}{\delta(K-1)!}}{2n}} \tag{41}$$

Hence, with probability at least $1 - \delta$ it holds that

$$R(h) - \widehat{R}_S(h) \leq \varepsilon = M^2\sqrt{\frac{K\log B + \log\frac{1}{\delta(K-1)!}}{2n}} \tag{42}$$

$\qquad\square$

### E.4.1. Proof of Theorem D.6

Observe that

$$|\mathcal{H}| = \sum_{j=1}^{B} \binom{B}{j} \le B \binom{B}{B/2} = \frac{B}{(B/2)!} B(B-1)\cdots(B-(K-1)) \le \frac{2B^{B/2}}{(B/2-1)!} \tag{43}$$

where we used the assumption that $B/2 \in \mathbb{N}$ to get $\frac{B}{(B/2)!} = \frac{B/2}{(B/2)\cdots 3} = \frac{2}{(B/2-1)!}$.

Therefore, $\log(|\mathcal{H}|) \le \log(2) + \frac{B}{2}\log(B) - \log((B/2-1)!)$

Now, let $S := \sum_{i=1}^{n}(h(x_i) - y_i)^2$, where by assumption, for all $i \in \{1,\ldots,n\}$ and any $h \in \mathcal{H}_{BSF}$, $(h(x_i) - y_i)^2 \le M^2$ with probability one. Then, by Hoeffding's inequality, for any $\varepsilon > 0$ and any $h \in \mathcal{H}_{BSF}$,

$$\mathbb{P}(\mathbb{E}(S) - S > n\varepsilon) = \mathbb{P}(R(h) - \widehat{R}_S(h) > \varepsilon) \le \exp\left(\frac{-2n\varepsilon^2}{M^4}\right) \tag{44}$$

Then, by the union bound, we have

$$\mathbb{P}(\exists h \in \mathcal{H}_{BSF} : R(h) - \widehat{R}_S(h) > \varepsilon) \le \sum_{h \in \mathcal{H}_{BSF}} \mathbb{P}(R(h) - \widehat{R}_S(h) > \varepsilon) \le \frac{2B^{B/2}}{(B/2-1)!}\exp\left(\frac{-2n\varepsilon^2}{M^4}\right) \tag{45}$$

Set $\frac{2B^{B/2}}{(B/2-1)!}\exp\left(\frac{-2n\varepsilon^2}{M^4}\right) = \delta$ and solve for $\varepsilon$ to get

$$\varepsilon = M^2 \sqrt{\frac{\frac{B}{2}\log B + \log\frac{1}{\delta(B/2-1)!} + \log(2)}{2n}} \tag{46}$$

Hence, with probability at least $1 - \delta$ it holds that

$$R(h) - \widehat{R}_S(h) \le \varepsilon = M^2 \sqrt{\frac{\frac{B}{2}\log B + \log\frac{1}{\delta(B/2-1)!} + \log(2)}{2n}} \tag{47}$$

$\square$

## F. Generalization bounds, simulation results

Table 1: Bound breach frequency, bound utilization and test-train risk discrepancy for different generalization bounds under different sample sizes with 100 trees

| Bound | $n = 0.35 \times 10^3, B = 100$ | | | $n = 0.35 \times 10^5, B = 100$ | | |
|---|---|---|---|---|---|---|
| | Breach % | Use % | Risk $\Delta\%$ | Breach % | Use % | Risk $\Delta\%$ |
| LASSO | 0 | $0.6 \pm 0.1$ | $30.8 \pm 17.2$ | 0 | $7.7 \pm 0.2$ | $0.1 \pm 1.0$ |
| BSF | 0 | $7.2 \pm 0.6$ | $16.5 \pm 11.9$ | 0 | $44.5 \pm 0.4$ | $0.1 \pm 1.0$ |
| SFS | 0 | $1.9 \pm 0.2$ | $11.9 \pm 12.7$ | 0 | $20.9 \pm 0.2$ | $0.0 \pm 1.0$ |

Table 2: Bound breach frequency, bound utilization and test-train risk discrepancy for different generalization bounds under different sample sizes with 500 trees

| Bound | $n = 0.35 \times 10^3, B = 500$ | | | $n = 0.35 \times 10^5, B = 500$ | | |
|---|---|---|---|---|---|---|
| | Breach % | Use % | Risk $\Delta\%$ | Breach % | Use % | Risk $\Delta\%$ |
| LASSO | 0 | $0.4 \pm 0.0$ | $77.2 \pm 26.7$ | 0 | $6.0 \pm 0.2$ | $0.1 \pm 1.0$ |
| BSF | 0 | $6.4 \pm 0.5$ | $25.0 \pm 12.7$ | 0 | $41.1 \pm 0.4$ | $0.2 \pm 1.2$ |
| SFS | 0 | $1.9 \pm 0.2$ | $22.8 \pm 13.6$ | 0 | $20.9 \pm 0.2$ | $0.4 \pm 1.3$ |

# G. Additional experiments

## G.1. Results with synthetic data

In the tables below, MSPE$\Delta \leq 0$ and #trees $\Delta \leq 0$ represent the empirical relative frequencies of said events, respectively (expressed as percentages). Note that by the Strong Law of Large Numbers, those empirical quantities converge almost surely to the true probabilities of observing said events (under weak standard assumptions). Hence, those quantities offer a complementary perspective to that provided by the more traditional significance tests for equality of means that we have included as well. Tables comparing the full and pruned forests consider a Bonferroni-adjusted significance level of $5\%/5 = 1\%$.

Table 3: Tree selection methods vs full forest, scenario 1: $n = 600$, 2 variables, $B = 25$, $\sigma_\varepsilon = 0.2$

| Method | Avg MSPE | Avg #trees | MSPE$\Delta$ vs FF | p-val | MSPE$\Delta \leq 0$ | Time (s) |
|---|---|---|---|---|---|---|
| Full Forest | 0.28 | 25.00 | - | - | - | - |
| SBS' | 0.27 | 3.70 | -4.8% | 0.05 | 55% | 0.01 |
| SFS | 0.21 | 7.56 | **-26.2%** | $10^{-16}$ | 89% | 0.01 |
| BSF | 0.21 | 3.95 | **-25.3%** | $10^{-16}$ | 90% | 0.20 |
| LASSO4 | 0.21 | 4.00 | **-26.3%** | $10^{-15}$ | 91% | 0.34 |
| LASSO | 0.18 | 11.12 | **-35.6%** | $10^{-16}$ | 97% | 0.29 |

Table 4: Comparing tree selection methods, scenario 1: $n = 600$, 2 variables, $B = 25$, $\sigma_\varepsilon = 0.2$

| Comparison | MSPE$\Delta$ | p-val | MSPE$\Delta \leq 0$ | #trees $\Delta$ | p-val | #trees $\Delta \leq 0$ |
|---|---|---|---|---|---|---|
| SBS' vs SFS | **+29.0%** | $10^{-16}$ | 7% | **-51.1%** | $10^{-9}$ | 91% |
| SBS' vs BSF | **+27.4%** | $10^{-16}$ | 3% | **-6.3%** | $10^{-10}$ | 91% |
| SBS' vs LASSO4 | **+29.1%** | $10^{-16}$ | 2% | **-7.5%** | $10^{-10}$ | 91% |
| SBS' vs LASSO | **+47.9%** | $10^{-16}$ | 0% | **-66.7%** | $10^{-11}$ | 92% |
| SFS vs BSF | **-1.2%** | 0.03 | 66% | **+91.4%** | $10^{-15}$ | 14% |
| SFS vs LASSO4 | +0.1% | 0.28 | 57% | **+89.0%** | $10^{-14}$ | 14% |
| SFS vs LASSO | **+14.7%** | $10^{-14}$ | 13% | **-32.0%** | $10^{-14}$ | 88% |
| BSF vs LASSO4 | +1.3% | 0.36 | 45% | **-1.3%** | 0.04 | 100% |
| BSF vs LASSO | **+16.1%** | $10^{-15}$ | 12% | **-64.5%** | $10^{-16}$ | 100% |
| LASSO4 vs LASSO | **+14.6%** | $10^{-16}$ | 4% | **-64.0%** | $10^{-16}$ | 100% |

Table 5: Tree selection methods vs full forest, scenario 2: $n = 20000$, 2 variables, $B = 25$, $\sigma_\varepsilon = 0.2$

| Method | Avg MSPE | Avg #trees | MSPE$\Delta$ vs FF | p-val | MSPE$\Delta \leq 0$ | Time (s) |
|---|---|---|---|---|---|---|
| Full Forest | 0.31 | 25.00 | - | - | - | - |
| SBS' | 0.24 | 3.59 | **-22.3%** | $10^{-16}$ | 98% | 0.25 |
| SFS | 0.21 | 7.91 | **-30.6%** | $10^{-16}$ | 100% | 0.26 |
| BSF | 0.21 | 4.00 | **-30.5%** | $10^{-16}$ | 100% | 3.54 |
| LASSO4 | 0.13 | 4.00 | **-58.8%** | $10^{-16}$ | 100% | 0.44 |
| LASSO | 0.11 | 11.37 | **-62.6%** | $10^{-16}$ | 100% | 0.43 |

Table 6: Comparing tree selection methods, scenario 2: $n = 20000$, 2 variables, $B = 25$, $\sigma_\varepsilon = 0.2$

| Comparison | MSPE$\Delta$ | p-val | MSPE$\Delta \leq 0$ | # trees $\Delta$ | p-val | # trees $\Delta \leq 0$ |
|---|---|---|---|---|---|---|
| SBS' vs SFS | **+12.0%** | $10^{-16}$ | 1% | **-54.6%** | $10^{-10}$ | 93% |
| SBS' vs BSF | **+11.9%** | $10^{-16}$ | 0% | **-10.3%** | $10^{-11}$ | 93% |
| SBS' vs LASSO4 | **+88.8%** | $10^{-16}$ | 0% | **-10.3%** | $10^{-11}$ | 93% |
| SBS' vs LASSO | **+108%** | $10^{-16}$ | 0% | **-68.4%** | $10^{-12}$ | 93% |
| SFS vs BSF | -0.1% | 0.05 | 65% | **+97.8%** | $10^{-16}$ | 10% |
| SFS vs LASSO4 | **+68.6%** | $10^{-16}$ | 2% | **+97.8%** | $10^{-16}$ | 10% |
| SFS vs LASSO | **+85.7%** | $10^{-16}$ | 0% | **-30.4%** | $10^{-15}$ | 89% |
| BSF vs LASSO4 | **+68.8%** | $10^{-16}$ | 2% | -0.0% | 1.00 | 100% |
| BSF vs LASSO | **+85.9%** | $10^{-16}$ | 0% | **-64.8%** | $10^{-16}$ | 100% |
| LASSO4 vs LASSO | **+10.2%** | $10^{-16}$ | 0% | **-64.8%** | $10^{-16}$ | 100% |

Table 7: Tree selection methods vs full forest, scenario 3: $n = 600$, 8 variables, $B = 25$, $\sigma_\varepsilon = 0.2$

| Method | Avg MSPE | Avg # trees | MSPE$\Delta$ vs FF | p-val | MSPE$\Delta \leq 0$ | Time (s) |
|---|---|---|---|---|---|---|
| Full Forest | 3.00 | 25.00 | - | - | - | - |
| SBS' | 3.45 | 6.63 | **+14.7%** | $10^{-16}$ | 11% | 0.01 |
| SFS | 3.16 | 9.01 | **+5.0%** | $10^{-7}$ | 29% | 0.01 |
| BSF | 3.49 | 4.00 | **+16.0%** | $10^{-16}$ | 3% | 0.19 |
| LASSO4 | 3.29 | 4.00 | **+9.4%** | $10^{-8}$ | 22% | 0.28 |
| LASSO | 2.31 | 11.98 | **-23.3%** | $10^{-16}$ | 95% | 0.28 |

Table 8: Comparing tree selection methods, scenario 3: $n = 600$, 8 variables, $B = 25$, $\sigma_\varepsilon = 0.2$

| Comparison | MSPE$\Delta$ | p-val | MSPE$\Delta \leq 0$ | # trees $\Delta$ | p-val | # trees $\Delta \leq 0$ |
|---|---|---|---|---|---|---|
| SBS' vs SFS | **+9.2%** | $10^{-14}$ | 16% | **-26.4%** | $10^{-6}$ | 79% |
| SBS' vs BSF | -1.1% | 0.40 | 53% | **+65.8%** | $10^{-6}$ | 46% |
| SBS' vs LASSO4 | **+4.9%** | $10^{-4}$ | 37% | **+65.8%** | $10^{-6}$ | 46% |
| SBS' vs LASSO | **+49.5%** | $10^{-16}$ | 0% | **-44.7%** | $10^{-13}$ | 84% |
| SFS vs BSF | **-9.4%** | $10^{-16}$ | 93% | **+125%** | $10^{-16}$ | 2% |
| SFS vs LASSO4 | **-4.0%** | $10^{-3}$ | 65% | **+125%** | $10^{-16}$ | 2% |
| SFS vs LASSO | **+36.9%** | $10^{-16}$ | 0% | **-24.8%** | $10^{-15}$ | 91% |
| BSF vs LASSO4 | **+6.0%** | $10^{-6}$ | 27% | 0.0% | 1.00 | 100% |
| BSF vs LASSO | **+51.1%** | $10^{-16}$ | 0% | **-66.6%** | $10^{-16}$ | 100% |
| LASSO4 vs LASSO | **+42.5%** | $10^{-16}$ | 0% | **-66.6%** | $10^{-16}$ | 100% |

Table 9: Tree selection methods vs full forest, scenario 4: $n = 20000$, 8 variables, $B = 25$, $\sigma_\varepsilon = 0.2$

| Method | Avg MSPE | Avg # trees | MSPE$\Delta$ vs FF | p-val | MSPE$\Delta \leq 0$ | Time (s) |
|---|---|---|---|---|---|---|
| Full Forest | 4.21 | 25.00 | - | - | - | - |
| SBS' | 4.16 | 9.91 | **-1.3%** | $10^{-9}$ | 71% | 0.26 |
| SFS | 4.09 | 11.27 | **-2.9%** | $10^{-16}$ | 97% | 0.26 |
| BSF | 4.24 | 4.00 | **+0.5%** | $10^{-3}$ | 37% | 3.00 |
| LASSO4 | 3.38 | 4.00 | **-19.9%** | $10^{-16}$ | 100% | 0.46 |
| LASSO | 2.53 | 12.00 | **-39.9%** | $10^{-16}$ | 100% | 0.46 |

Table 10: Comparing tree selection methods, scenario 4: $n = 20000$, 8 variables, $B = 25$, $\sigma_\varepsilon = 0.2$

| Comparison | MSPE$\Delta$ | p-val | MSPE$\Delta \leq 0$ | # trees $\Delta$ | p-val | # trees $\Delta \leq 0$ |
|---|---|---|---|---|---|---|
| SBS' vs SFS | **+1.6%** | $10^{-16}$ | 0% | **-12.1%** | $10^{-4}$ | 77% |
| SBS' vs BSF | **-1.8%** | $10^{-6}$ | 91% | **+148%** | $10^{-16}$ | 2% |
| SBS' vs LASSO4 | **+23.2%** | $10^{-16}$ | 0% | **+148%** | $10^{-16}$ | 2% |
| SBS' vs LASSO | **+64.2%** | $10^{-16}$ | 0% | **-17.4%** | $10^{-5}$ | 78% |
| SFS vs BSF | **-3.4%** | $10^{-16}$ | 100% | **+182%** | $10^{-16}$ | 0% |
| SFS vs LASSO4 | **+21.3%** | $10^{-16}$ | 0% | **+182%** | $10^{-16}$ | 0% |
| SFS vs LASSO | **+61.7%** | $10^{-16}$ | 0% | **-6.1%** | $10^{-4}$ | 76% |
| BSF vs LASSO4 | **+25.5%** | $10^{-16}$ | 0% | 0.0% | 1.00 | 100% |
| BSF vs LASSO | **+67.3%** | $10^{-16}$ | 0% | **-66.7%** | $10^{-16}$ | 100% |
| LASSO4 vs LASSO | **+33.3%** | $10^{-16}$ | 0% | **-66.7%** | $10^{-16}$ | 100% |

Table 11: Tree selection methods vs full forest, scenario 5: $n = 600$, 2 variables, $B = 100$, $\sigma_\varepsilon = 0.2$

| Method | Avg MSPE | Avg # trees | MSPE$\Delta$ vs FF | p-val | MSPE$\Delta \leq 0$ | Time (s) |
|---|---|---|---|---|---|---|
| Full Forest | 0.29 | 100.00 | - | - | - | - |
| SBS' | 0.26 | 7.38 | **-9.7%** | $10^{-7}$ | 74% | 0.45 |
| SFS | 0.18 | 13.10 | **-36.7%** | $10^{-16}$ | 99% | 0.42 |
| BSF | 0.20 | 3.98 | **-30.5%** | $10^{-16}$ | 98% | 49.58 |
| LASSO4 | 0.20 | 4.00 | **-30.0%** | $10^{-16}$ | 95% | 0.64 |
| LASSO | 0.16 | 20.44 | **-45.2%** | $10^{-16}$ | 100% | 0.33 |

Table 12: Comparing tree selection methods, scenario 5: $n = 600$, 2 variables, $B = 100$, $\sigma_\varepsilon = 0.2$

| Comparison | MSPE$\Delta$ | p-val | MSPE$\Delta \leq 0$ | # trees $\Delta$ | p-val | # trees $\Delta \leq 0$ |
|---|---|---|---|---|---|---|
| SBS' vs SFS | **+42.7%** | $10^{-16}$ | 2% | **-43.7%** | $10^{-11}$ | 94% |
| SBS' vs BSF | **+29.9%** | $10^{-16}$ | 3% | **+85.4%** | $10^{-12}$ | 93% |
| SBS' vs LASSO4 | **+28.9%** | $10^{-16}$ | 9% | **+84.5%** | $10^{-12}$ | 93% |
| SBS' vs LASSO | **+64.9%** | $10^{-16}$ | 1% | **-63.9%** | $10^{-11}$ | 94% |
| SFS vs BSF | **-9.0%** | $10^{-12}$ | 81% | **+229%** | $10^{-16}$ | 1% |
| SFS vs LASSO4 | **-9.6%** | $10^{-7}$ | 73% | **+228%** | $10^{-16}$ | 1% |
| SFS vs LASSO | **+15.6%** | $10^{-14}$ | 15% | **-35.9%** | $10^{-14}$ | 86% |
| BSF vs LASSO4 | -0.7% | 0.71 | 49% | -0.5% | 0.35 | 100% |
| BSF vs LASSO | **+27.0%** | $10^{-16}$ | 3% | **-80.5%** | $10^{-16}$ | 100% |
| LASSO4 vs LASSO | **+27.9%** | $10^{-16}$ | 4% | **-80.4%** | $10^{-16}$ | 100% |

Table 13: Tree selection methods vs full forest, scenario 6: $n = 20000$, 2 variables, $B = 100$, $\sigma_\varepsilon = 0.2$

| Method | Avg MSPE | Avg # trees | MSPE$\Delta$ vs FF | p-val | MSPE$\Delta \leq 0$ | Time (s) |
|---|---|---|---|---|---|---|
| Full Forest | 0.29 | 100.00 | - | - | - | - |
| SBS' | 0.23 | 2.00 | **-19.7%** | $10^{-16}$ | 99% | 13.37 |
| SFS | 0.19 | 13.59 | **-33.7%** | $10^{-16}$ | 100% | 10.83 |
| BSF | 0.20 | 4.00 | **-31.8%** | $10^{-16}$ | 100% | 872.6 |
| LASSO4 | 0.11 | 4.00 | **-61.8%** | $10^{-16}$ | 100% | 0.90 |
| LASSO | 0.08 | 20.11 | **-71.4%** | $10^{-16}$ | 100% | 0.43 |

Table 14: Comparing tree selection methods, scenario 6: $n = 20000$, 2 variables, $B = 100$, $\sigma_\varepsilon = 0.2$

| Comparison | MSPE$\Delta$ | p-val | MSPE$\Delta \leq 0$ | # trees $\Delta$ | p-val | # trees $\Delta \leq 0$ |
|---|---|---|---|---|---|---|
| SBS' vs SFS | **+21.0%** | $10^{-16}$ | 0% | **-85.3%** | $10^{-16}$ | 100% |
| SBS' vs BSF | **+17.7%** | $10^{-16}$ | 0% | **-50%** | $10^{-16}$ | 100% |
| SBS' vs LASSO4 | **+110%** | $10^{-12}$ | 0% | **-50%** | $10^{-16}$ | 100% |
| SBS' vs LASSO | **+180%** | $10^{-16}$ | 0% | **-90.1%** | $10^{-16}$ | 100% |
| SFS vs BSF | **-2.8%** | $10^{-16}$ | 92% | **+240%** | $10^{-16}$ | 0% |
| SFS vs LASSO4 | **+73.5%** | $10^{-16}$ | 0% | **+240%** | $10^{-16}$ | 0% |
| SFS vs LASSO | **+132%** | $10^{-16}$ | 0% | **-32.4%** | $10^{-14}$ | 88% |
| BSF vs LASSO4 | **+78.5%** | $10^{-16}$ | 0% | 0.0% | 1.00 | 100% |
| BSF vs LASSO | **+138%** | $10^{-16}$ | 0% | **-80.1%** | $10^{-16}$ | 100% |
| LASSO4 vs LASSO | **+33.5%** | $10^{-16}$ | 0% | **-80.1%** | $10^{-16}$ | 100% |

Table 15: Tree selection methods vs full forest, scenario 7: $n = 600$, 8 variables, $B = 100$, $\sigma_\varepsilon = 0.2$

| Method | Avg MSPE | Avg # trees | MSPE$\Delta$ vs FF | p-val | MSPE$\Delta \leq 0$ | Time (s) |
|---|---|---|---|---|---|---|
| Full Forest | 2.89 | 100.00 | - | - | - | - |
| SBS' | 3.50 | 5.52 | **+20.8%** | $10^{-15}$ | 5% | 0.51 |
| SFS | 2.92 | 14.20 | +1.1% | 0.36 | 49% | 0.42 |
| BSF | 3.36 | 4.00 | **+16.2%** | $10^{-16}$ | 5% | 50.54 |
| LASSO4 | 3.20 | 4.00 | **+10.6%** | $10^{-10}$ | 20% | 0.72 |
| LASSO | 1.61 | 32.31 | **-44.5%** | $10^{-16}$ | 100% | 0.34 |

Table 16: Comparing tree selection methods, scenario 7: $n = 600$, 8 variables, $B = 100$, $\sigma_\varepsilon = 0.2$

| Comparison | MSPE$\Delta$ | p-val | MSPE$\Delta \leq 0$ | # trees $\Delta$ | p-val | # trees $\Delta \leq 0$ |
|---|---|---|---|---|---|---|
| SBS' vs SFS | **+19.6%** | $10^{-16}$ | 5% | **-61.1%** | $10^{-15}$ | 98% |
| SBS' vs BSF | **+4.0%** | $10^{-4}$ | 35% | **+38.0%** | 0.05 | 67% |
| SBS' vs LASSO4 | **+9.2%** | $10^{-6}$ | 30% | **+38.0%** | 0.05 | 67% |
| SBS' vs LASSO | **+118%** | $10^{-16}$ | 0% | **-82.9%** | $10^{-16}$ | 98% |
| SFS vs BSF | **-13.0%** | $10^{-16}$ | 96% | **+255%** | $10^{-16}$ | 0% |
| SFS vs LASSO4 | **-8.6%** | $10^{-8}$ | 74% | **+255%** | $10^{-16}$ | 0% |
| SFS vs LASSO | **+82.0%** | $10^{-16}$ | 0% | **-56.1%** | $10^{-16}$ | 100% |
| BSF vs LASSO4 | **+5.0%** | $10^{-4}$ | 34% | **0.0%** | 1.00 | 100% |
| BSF vs LASSO | **+109%** | $10^{-16}$ | 0% | **-87.6%** | $10^{-16}$ | 100% |
| LASSO4 vs LASSO | **+99.2%** | $10^{-16}$ | 0% | **-87.6%** | $10^{-16}$ | 100% |

Table 17: Tree selection methods vs full forest, scenario 8: $n = 20000$, 8 variables, $B = 100$, $\sigma_\varepsilon = 0.2$

| Method | Avg MSPE | Avg # trees | MSPE$\Delta$ vs FF | p-val | MSPE$\Delta \leq 0$ | Time (s) |
|---|---|---|---|---|---|---|
| Full Forest | 4.15 | 100.00 | - | - | - | - |
| SBS' | 4.09 | 10.26 | **-1.6%** | $10^{-15}$ | 88% | 13.05 |
| SFS | 3.95 | 19.49 | **-4.9%** | $10^{-16}$ | 100% | 10.83 |
| BSF | 4.14 | 4.00 | **-0.4%** | 0.01 | 61% | 890.1 |
| LASSO4 | 3.36 | 4.00 | **-19.1%** | $10^{-16}$ | 100% | 0.86 |
| LASSO | 2.05 | 45.79 | **-50.7%** | $10^{-16}$ | 100% | 0.88 |

Table 18: Comparing tree selection methods, scenario 8: $n = 20000$, 8 variables, $B = 100$, $\sigma_\varepsilon = 0.2$

| Comparison | MSPE$\Delta$ | p-val | MSPE$\Delta \leq 0$ | # trees $\Delta$ | p-val | # trees $\Delta \leq 0$ |
|---|---|---|---|---|---|---|
| SBS' vs SFS | **+3.5%** | $10^{-16}$ | 0% | **-47.4%** | $10^{-15}$ | 91% |
| SBS' vs BSF | **-1.2%** | $10^{-11}$ | 80% | **+157%** | $10^{-16}$ | 2% |
| SBS' vs LASSO4 | **+21.7%** | $10^{-16}$ | 0% | **+157%** | $10^{-16}$ | 2% |
| SBS' vs LASSO | **+99.7%** | $10^{-16}$ | 0% | **-77.6%** | $10^{-16}$ | 99% |
| SFS vs BSF | **-4.5%** | $10^{-16}$ | 100% | **+387%** | $10^{-16}$ | 0% |
| SFS vs LASSO4 | **+17.6%** | $10^{-16}$ | 0% | **+387%** | $10^{-16}$ | 0% |
| SFS vs LASSO | **+92.9%** | $10^{-16}$ | 0% | **-57.4%** | $10^{-16}$ | 100% |
| BSF vs LASSO4 | **+23.2%** | $10^{-16}$ | 0% | **0.0%** | 1.00 | 100% |
| BSF vs LASSO | **+102%** | $10^{-16}$ | 0% | **-91.3%** | $10^{-16}$ | 100% |
| LASSO4 vs LASSO | **+64.1%** | $10^{-16}$ | 0% | **-91.3%** | $10^{-16}$ | 100% |

Table 19: Tree selection methods vs full forest, scenario 9: $n = 600$, 2 variables, $B = 25$, $\sigma_\varepsilon = \sqrt{2}$

| Method | Avg MSPE | Avg # trees | MSPE$\Delta$ vs FF | p-val | MSPE$\Delta \leq 0$ | Time (s) |
|---|---|---|---|---|---|---|
| Full Forest | 2.36 | 25.00 | - | - | - | - |
| SBS' | 2.64 | 4.16 | **+11.7%** | $10^{-16}$ | 7% | 0.01 |
| SFS | 2.46 | 6.83 | **+4.4%** | $10^{-7}$ | 26% | 0.01 |
| BSF | 2.51 | 3.92 | **+6.2%** | $10^{-13}$ | 20% | 0.18 |
| LASSO4 | 2.74 | 3.85 | **+16.2%** | $10^{-16}$ | 3% | 0.30 |
| LASSO | 2.73 | 5.82 | **+15.7%** | $10^{-16}$ | 4% | 0.15 |

Table 20: Comparing Tree Selection Methods, Scenario 9: $n = 600$, 2 Variables, $B = 25$, $\sigma_\varepsilon = \sqrt{2}$

| Comparison | MSPE$\Delta$ | p-val | MSPE$\Delta \leq 0$ | # trees $\Delta$ | p-val | # trees $\Delta \leq 0$ |
|---|---|---|---|---|---|---|
| SBS' vs SFS | **+7.0%** | $10^{-12}$ | 21% | **-39.1%** | $10^{-9}$ | 85% |
| SBS' vs BSF | **+5.2%** | $10^{-8}$ | 25% | **+6.1%** | 0.02 | 74% |
| SBS' vs LASSO4 | **-3.9%** | $10^{-3}$ | 60% | **+8.1%** | 0.02 | 74% |
| SBS' vs LASSO | **-3.4%** | 0.02 | 56% | **-28.5%** | $10^{-6}$ | 79% |
| SFS vs BSF | **-1.7%** | $10^{-5}$ | 73% | **+74.2%** | $10^{-14}$ | 17% |
| SFS vs LASSO4 | **-10.2%** | $10^{-15}$ | 89% | **+77.4%** | $10^{-14}$ | 16% |
| SFS vs LASSO | **-9.8%** | $10^{-14}$ | 87% | **+17.4%** | $10^{-3}$ | 42% |
| BSF vs LASSO4 | **-8.6%** | $10^{-15}$ | 86% | +1.8% | 0.15 | 92% |
| BSF vs LASSO | **-8.2%** | $10^{-13}$ | 85% | **-32.6%** | $10^{-12}$ | 91% |
| LASSO4 vs LASSO | +0.4% | 0.10 | 39% | **-33.8%** | $10^{-13}$ | 6% |

Table 21: Tree selection methods vs full forest, scenario 10: $n = 20000$, 2 variables, $B = 25$, $\sigma_\varepsilon = \sqrt{2}$

| Method | Avg MSPE | Avg # trees | MSPE$\Delta$ vs FF | p-val | MSPE$\Delta \leq 0$ | Time (s) |
|---|---|---|---|---|---|---|
| Full Forest | 2.31 | 25.00 | - | - | - | - |
| SBS' | 2.24 | 2.34 | **-2.6%** | $10^{-16}$ | 88% | 0.22 |
| SFS | 2.20 | 7.66 | **-4.7%** | $10^{-16}$ | 100% | 0.23 |
| BSF | 2.20 | 3.98 | **-4.5%** | $10^{-16}$ | 100% | 3.28 |
| LASSO4 | 2.19 | 4.00 | **-5.1%** | $10^{-16}$ | 100% | 0.50 |
| LASSO | 2.17 | 11.05 | **-6.1%** | $10^{-16}$ | 100% | 0.50 |

Table 22: Comparing tree selection methods, scenario 10: $n = 20000$, 2 variables, $B = 25$, $\sigma_\varepsilon = \sqrt{2}$

| Comparison | MSPE$\Delta$ | p-val | MSPE$\Delta \leq 0$ | # trees $\Delta$ | p-val | # trees $\Delta \leq 0$ |
|---|---|---|---|---|---|---|
| SBS' vs SFS | **+2.1%** | $10^{-16}$ | 4% | **-68.5%** | $10^{-16}$ | 98% |
| SBS' vs BSF | **+1.9%** | $10^{-16}$ | 0% | **-41.2%** | $10^{-16}$ | 99% |
| SBS' vs LASSO4 | **+2.5%** | $10^{-16}$ | 3% | **-41.4%** | $10^{-16}$ | 99% |
| SBS' vs LASSO | **+3.6%** | $10^{-16}$ | 0% | **-78.8%** | $10^{-16}$ | 99% |
| SFS vs BSF | **-0.2%** | $10^{-6}$ | 77% | **+92.5%** | $10^{-16}$ | 9% |
| SFS vs LASSO4 | +0.4% | 0.11 | 52% | **+92.0%** | $10^{-16}$ | 9% |
| SFS vs LASSO | **+1.5%** | $10^{-14}$ | 16% | **-30.7%** | $10^{-16}$ | 96% |
| BSF vs LASSO4 | +0.6% | $10^{-3}$ | 45% | -0.3% | 0.78 | 99% |
| BSF vs LASSO | **+1.7%** | $10^{-16}$ | 10% | **-64.0%** | $10^{-16}$ | 100% |
| LASSO4 vs LASSO | **+1.1%** | $10^{-14}$ | 14% | **-63.9%** | $10^{-16}$ | 100% |

Table 23: Tree selection methods vs full forest, scenario 11: $n = 600$, 8 variables, $B = 25$, $\sigma_\varepsilon = \sqrt{2}$

| Method | Avg MSPE | Avg # trees | MSPE$\Delta$ vs FF | p-val | MSPE$\Delta \leq 0$ | Time (s) |
|---|---|---|---|---|---|---|
| Full Forest | 4.48 | 25.00 | - | - | - | - |
| SBS' | 5.11 | 5.93 | **+14.0%** | $10^{-16}$ | 9% | 0.01 |
| SFS | 4.73 | 8.18 | **+5.5%** | $10^{-9}$ | 26% | 0.01 |
| BSF | 5.10 | 4.00 | **+13.8%** | $10^{-16}$ | 8% | 0.19 |
| LASSO4 | 5.25 | 4.00 | **+17.1%** | $10^{-16}$ | 4% | 0.25 |
| LASSO | 4.60 | 11.44 | +2.7% | 0.01 | 36% | 0.15 |

Table 24: Comparing tree selection methods, scenario 11: $n = 600$, 8 variables, $B = 25$, $\sigma_\varepsilon = \sqrt{2}$

| Comparison | MSPE$\Delta$ | p-val | MSPE$\Delta \leq 0$ | # trees $\Delta$ | p-val | # trees $\Delta \leq 0$ |
|---|---|---|---|---|---|---|
| SBS' vs SFS | **+8.0%** | $10^{-13}$ | 18% | **-27.5%** | $10^{-7}$ | 81% |
| SBS' vs BSF | +0.2% | 0.93 | 48% | **+48.3%** | $10^{-3}$ | 57% |
| SBS' vs LASSO4 | **-2.7%** | 0.02 | 59% | **+48.3%** | $10^{-3}$ | 57% |
| SBS' vs LASSO | **+11.0%** | $10^{-13}$ | 16% | **-48.2%** | $10^{-13}$ | 90% |
| SFS vs BSF | **-7.2%** | $10^{15}$ | 91% | **+105%** | $10^{-16}$ | 4% |
| SFS vs LASSO4 | **-9.9%** | $10^{-15}$ | 92% | **+105%** | $10^{-16}$ | 4% |
| SFS vs LASSO | **+2.8%** | $10^{-3}$ | 35% | **-28.5%** | $10^{-16}$ | 97% |
| BSF vs LASSO4 | **-2.8%** | $10^{-5}$ | 68% | -0.0% | 1.00 | 100% |
| BSF vs LASSO | **+10.8%** | $10^{-14}$ | 12% | **-65.0%** | $10^{-16}$ | 100% |
| LASSO4 vs LASSO | **+14.1%** | $10^{-16}$ | 4% | **-65.0%** | $10^{-16}$ | 100% |

Table 25: Tree selection methods vs full forest, scenario 12: $n = 20000$, 8 variables, $B = 25$, $\sigma_\varepsilon = \sqrt{2}$

| Method | Avg MSPE | Avg # trees | MSPE$\Delta$ vs FF | p-val | MSPE$\Delta \leq 0$ | Time (s) |
|---|---|---|---|---|---|---|
| Full Forest | 6.34 | 25.00 | - | - | - | - |
| SBS' | 6.27 | 9.24 | **-1.1%** | $10^{-10}$ | 79% | 0.23 |
| SFS | 6.19 | 10.68 | **-2.4%** | $10^{-16}$ | 98% | 0.24 |
| BSF | 6.32 | 4.00 | -0.2% | 0.13 | 58% | 3.25 |
| LASSO4 | 5.45 | 4.00 | **-13.9%** | $10^{-16}$ | 100% | 0.47 |
| LASSO | 4.57 | 12.00 | **-27.9%** | $10^{-16}$ | 100% | 0.45 |

Table 26: Comparing tree selection methods, scenario 12: $n = 20000$, 8 variables, $B = 25$, $\sigma_\varepsilon = \sqrt{2}$

| Comparison | MSPE$\Delta$ | p-val | MSPE$\Delta \leq 0$ | # trees $\Delta$ | p-val | # trees $\Delta \leq 0$ |
|---|---|---|---|---|---|---|
| SBS' vs SFS | **+1.3%** | $10^{-16}$ | 1% | **-13.5%** | $10^{-4}$ | 72% |
| SBS' vs BSF | **-0.9%** | $10^{-12}$ | 88% | **+131%** | $10^{-16}$ | 5% |
| SBS' vs LASSO4 | **+14.9%** | $10^{-16}$ | 0% | **+131%** | $10^{-16}$ | 5% |
| SBS' vs LASSO | **+37.2%** | $10^{-16}$ | 0% | **-23.0%** | $10^{-6}$ | 79% |
| SFS vs BSF | **-2.2%** | $10^{-16}$ | 99% | **+167%** | $10^{-16}$ | 0% |
| SFS vs LASSO4 | **+13.4%** | $10^{-16}$ | 0% | **+167%** | $10^{-16}$ | 0% |
| SFS vs LASSO | **+35.4%** | $10^{-16}$ | 0% | **-11.0%** | $10^{-8}$ | 80% |
| BSF vs LASSO4 | **+15.9%** | $10^{-16}$ | 0% | 0.0% | 1.00 | 100% |
| BSF vs LASSO | **+38.4%** | $10^{-16}$ | 0% | **-66.7%** | $10^{-16}$ | 100% |
| LASSO4 vs LASSO | **+19.4%** | $10^{-16}$ | 0% | **-66.7%** | $10^{-16}$ | 100% |

Table 27: Tree selection methods vs full forest, scenario 13: $n = 600$, 2 variables, $B = 100$, $\sigma_\varepsilon = \sqrt{2}$

| Method | Avg MSPE | Avg # trees | MSPE$\Delta$ vs FF | p-val | MSPE$\Delta \leq 0$ | Time (s) |
|---|---|---|---|---|---|---|
| Full Forest | 2.32 | 100.00 | - | - | - | - |
| SBS' | 2.63 | 4.88 | **+13.6%** | $10^{-16}$ | 5% | 0.47 |
| SFS | 2.37 | 11.48 | **+2.1%** | $10^{-5}$ | 32% | 0.40 |
| BSF | 2.50 | 3.99 | **+7.8%** | $10^{-14}$ | 16% | 52.33 |
| LASSO4 | 2.70 | 3.91 | **+16.5%** | $10^{-16}$ | 4% | 1.02 |
| LASSO | 2.66 | 8.36 | **+14.8%** | $10^{-16}$ | 4% | 0.49 |

Table 28: Comparing tree selection methods, scenario 13: $n = 600$, 2 variables, $B = 100$, $\sigma_\varepsilon = \sqrt{2}$

| Comparison | MSPE$\Delta$ | p-val | MSPE$\Delta \leq 0$ | # trees $\Delta$ | p-val | # trees $\Delta \leq 0$ |
|---|---|---|---|---|---|---|
| SBS' vs SFS | **+11.2%** | $10^{-16}$ | 5% | **-58.3%** | $10^{-14}$ | 95% |
| SBS' vs BSF | **+5.3%** | $10^{-8}$ | 24% | **+22.3%** | $10^{-4}$ | 80% |
| SBS' vs LASSO4 | -2.5% | 0.13 | 53% | **+24.8%** | $10^{-4}$ | 79% |
| SBS' vs LASSO | -1.0% | 0.72 | 47% | **-41.6%** | $10^{-11}$ | 88% |
| SFS vs BSF | **-5.3%** | $10^{-14}$ | 88% | **+193%** | $10^{-16}$ | 1% |
| SFS vs LASSO4 | **-12.3%** | $10^{-16}$ | 94% | **+199%** | $10^{-16}$ | 1% |
| SFS vs LASSO | **-11.0%** | $10^{-16}$ | 93% | **+40.0%** | $10^{-10}$ | 33% |
| BSF vs LASSO4 | **-7.4%** | $10^{-11}$ | 81% | +2.0% | 0.10 | 95% |
| BSF vs LASSO | **-6.0%** | $10^{-8}$ | 76% | **-52.3%** | $10^{-16}$ | 97% |
| LASSO4 vs LASSO | **+1.5%** | $10^{-3}$ | 39% | **-53.2%** | $10^{-16}$ | 100% |

Table 29: Tree selection methods vs full forest, scenario 14: $n = 20000$, 2 variables, $B = 100$, $\sigma_\varepsilon = \sqrt{2}$

| Method | Avg MSPE | Avg # trees | MSPE$\Delta$ vs FF | p-val | MSPE$\Delta \leq 0$ | Time (s) |
|---|---|---|---|---|---|---|
| Full Forest | 2.29 | 100.00 | - | - | - | - |
| SBS' | 2.22 | 2.00 | **-2.5%** | $10^{-16}$ | 95% | 13.10 |
| SFS | 2.16 | 13.20 | **-5.6%** | $10^{-16}$ | 100% | 10.63 |
| BSF | 2.17 | 4.00 | **-5.1%** | $10^{-16}$ | 100% | 862.2 |
| LASSO4 | 2.15 | 3.97 | **-5.9%** | $10^{-16}$ | 100% | 1.01 |
| LASSO | 2.11 | 19.72 | **-7.6%** | $10^{-16}$ | 100% | 0.48 |

Table 30: Comparing tree selection methods, scenario 14: $n = 20000$, 2 variables, $B = 100$, $\sigma_\varepsilon = \sqrt{2}$

| Comparison | MSPE$\Delta$ | p-val | MSPE$\Delta \leq 0$ | # trees $\Delta$ | p-val | # trees $\Delta \leq 0$ |
|---|---|---|---|---|---|---|
| SBS' vs SFS | **+3.3%** | $10^{-16}$ | 1% | **-84.8%** | $10^{-16}$ | 100% |
| SBS' vs BSF | **+2.7%** | $10^{-16}$ | 0% | **-50.0%** | $10^{-16}$ | 100% |
| SBS' vs LASSO4 | **+3.6%** | $10^{-16}$ | 1% | **-49.6%** | $10^{-16}$ | 100% |
| SBS' vs LASSO | **+5.5%** | $10^{-16}$ | 0% | **-89.8%** | $10^{-16}$ | 100% |
| SFS vs BSF | **-0.6%** | $10^{-14}$ | 89% | **+230%** | $10^{-16}$ | 2% |
| SFS vs LASSO4 | +0.3% | 0.06 | 41% | **+232%** | $10^{-16}$ | 2% |
| SFS vs LASSO | **+2.1%** | $10^{-16}$ | 0% | **-33.1%** | $10^{-16}$ | 94% |
| BSF vs LASSO4 | **+0.9%** | $10^{-7}$ | 30% | +0.8% | 0.15 | 97% |
| BSF vs LASSO | **+2.7%** | $10^{-16}$ | 0% | **-79.7%** | $10^{-16}$ | 100% |
| LASSO4 vs LASSO | **+1.8%** | $10^{-16}$ | 7% | **-79.9%** | $10^{-16}$ | 100% |

Table 31: Tree selection methods vs full forest, scenario 15: $n = 600$, 8 variables, $B = 100$, $\sigma_\varepsilon = \sqrt{2}$

| Method | Avg MSPE | Avg # trees | MSPE$\Delta$ vs FF | p-val | MSPE$\Delta \leq 0$ | Time (s) |
|---|---|---|---|---|---|---|
| Full Forest | 4.45 | 100.00 | - | - | - | - |
| SBS' | 5.31 | 5.25 | **+19.3%** | $10^{-16}$ | 7% | 0.48 |
| SFS | 4.55 | 12.95 | **+2.4%** | $10^{-4}$ | 39% | 0.42 |
| BSF | 5.08 | 4.00 | **+14.4%** | $10^{-16}$ | 10% | 51.19 |
| LASSO4 | 5.20 | 4.00 | **+17.0%** | $10^{-16}$ | 2% | 0.83 |
| LASSO | 4.15 | 21.77 | **-6.6%** | $10^{-8}$ | 75% | 0.42 |

Table 32: Comparing tree selection methods, scenario 15: $n = 600$, 8 variables, $B = 100$, $\sigma_\varepsilon = \sqrt{2}$

| Comparison | MSPE$\Delta$ | p-val | MSPE$\Delta \leq 0$ | # trees $\Delta$ | p-val | # trees $\Delta \leq 0$ |
|---|---|---|---|---|---|---|
| SBS' vs SFS | **+16.6%** | $10^{-16}$ | 9% | **-59.5%** | $10^{-15}$ | 97% |
| SBS' vs BSF | **+4.3%** | $10^{-4}$ | 35% | +31.3% | 0.19 | 69% |
| SBS' vs LASSO4 | +2.0% | 0.14 | 45% | +31.3% | 0.19 | 69% |
| SBS' vs LASSO | **+27.8%** | $10^{-16}$ | 3% | **-75.9%** | $10^{-15}$ | 98% |
| SFS vs BSF | **-10.5%** | $10^{-16}$ | 91% | **+224%** | $10^{-16}$ | 0% |
| SFS vs LASSO4 | **-12.5%** | $10^{-16}$ | 96% | **+224%** | $10^{-16}$ | 0% |
| SFS vs LASSO | **+9.6%** | $10^{-13}$ | 13% | **-40.5%** | $10^{-16}$ | 100% |
| BSF vs LASSO4 | **-2.3%** | 0.01 | 60% | 0.0% | 1.00 | 100% |
| BSF vs LASSO | **+22.4%** | $10^{-16}$ | 4% | **-81.6%** | $10^{-16}$ | 100% |
| LASSO4 vs LASSO | **+25.3%** | $10^{-16}$ | 0% | **-81.6%** | $10^{-16}$ | 100% |

Table 33: Tree selection methods vs full forest, scenario 16: $n = 20000$, 8 variables, $B = 100$, $\sigma_\varepsilon = \sqrt{2}$

| Method | Avg MSPE | Avg # trees | MSPE$\Delta$ vs FF | p-val | MSPE$\Delta \leq 0$ | Time (s) |
|---|---|---|---|---|---|---|
| Full Forest | 6.29 | 100.00 | - | - | - | - |
| SBS' | 6.22 | 8.99 | **-1.1%** | $10^{-11}$ | 80% | 14.85 |
| SFS | 6.05 | 18.09 | **-3.7%** | $10^{-16}$ | 100% | 12.46 |
| BSF | 6.24 | 4.00 | **-0.7%** | $10^{-8}$ | 73% | 898.9 |
| LASSO4 | 5.42 | 4.00 | **-13.7%** | $10^{-16}$ | 100% | 0.86 |
| LASSO | 4.02 | 43.42 | **-36.0%** | $10^{-16}$ | 100% | 0.43 |

Table 34: Comparing tree selection methods, scenario 16: $n = 20000$, 8 variables, $B = 100$, $\sigma_\varepsilon = \sqrt{2}$

| Comparison | MSPE$\Delta$ | p-val | MSPE$\Delta \leq 0$ | # trees $\Delta$ | p-val | # trees $\Delta \leq 0$ |
|---|---|---|---|---|---|---|
| SBS' vs SFS | **+2.7%** | $10^{-16}$ | 0% | **-50.3%** | $10^{-14}$ | 94% |
| SBS' vs BSF | **-0.4%** | $10^{-5}$ | 70% | **+125%** | $10^{-16}$ | 6% |
| SBS' vs LASSO4 | **+14.6%** | $10^{-16}$ | 0% | **+125%** | $10^{-16}$ | 6% |
| SBS' vs LASSO | **+54.5%** | $10^{-16}$ | 0% | **-79.3%** | $10^{-16}$ | 98% |
| SFS vs BSF | **-3.0%** | $10^{-16}$ | 100% | **+352%** | $10^{-16}$ | 0% |
| SFS vs LASSO4 | **+11.6%** | $10^{-16}$ | 2% | **+352%** | $10^{-16}$ | 0% |
| SFS vs LASSO | **+50.4%** | $10^{-16}$ | 0% | **-58.3%** | $10^{-16}$ | 100% |
| BSF vs LASSO4 | **+15.1%** | $10^{-16}$ | 0% | 0.0% | 1.00 | 100% |
| BSF vs LASSO | **+55.1%** | $10^{-16}$ | 0% | **-90.8%** | $10^{-16}$ | 100% |
| LASSO4 vs LASSO | **+34.8%** | $10^{-16}$ | 0% | **-90.8%** | $10^{-16}$ | 100% |

## G.2. Graphical summary of performance results across synthetic scenarios

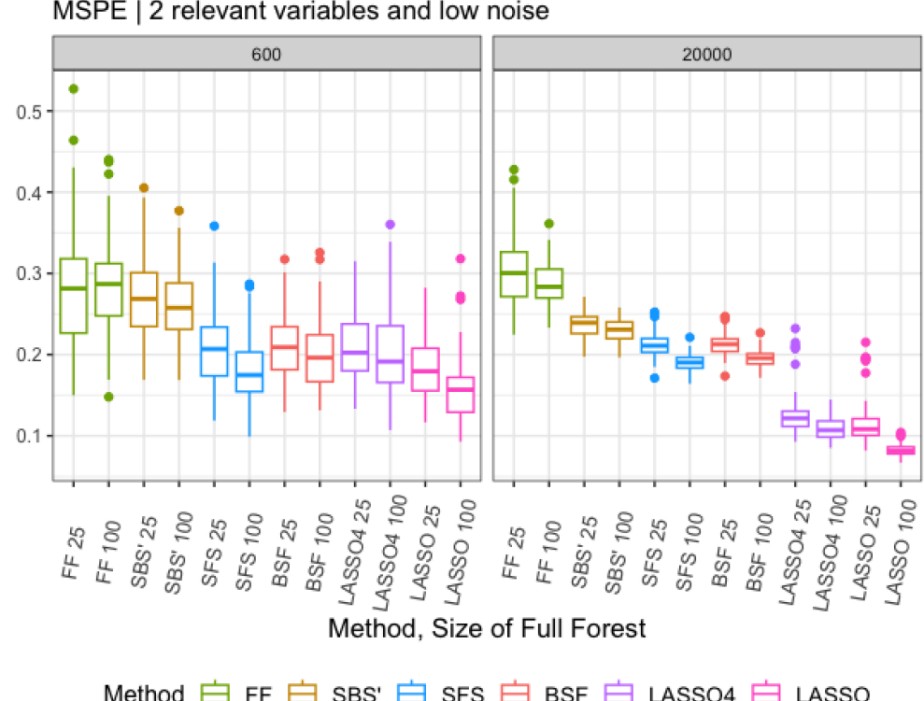

Figure 5: MSPE of full and pruned forests in scenarios with 2 relevant variables and low noise

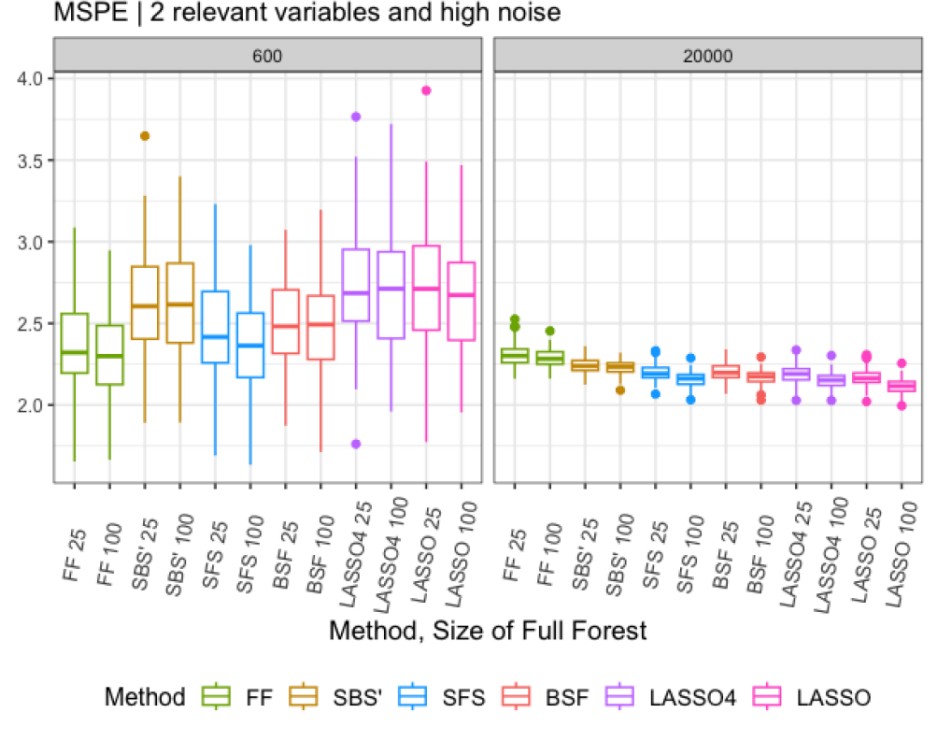

Figure 6: MSPE of full and pruned forests in scenarios with 2 relevant variables and high noise

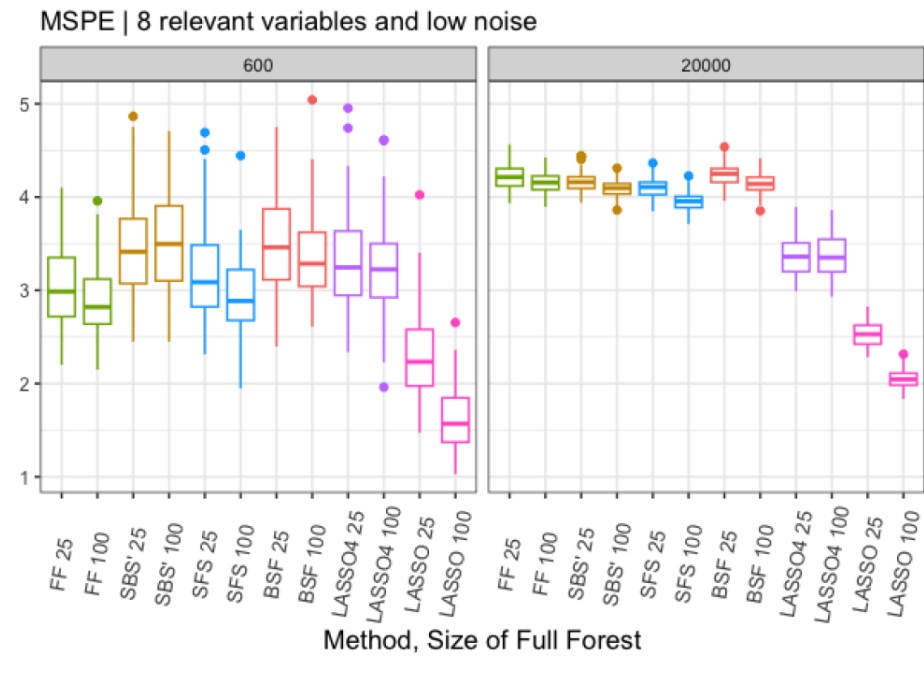

Figure 7: MSPE of full and pruned forests in scenarios with 8 relevant variables and low noise

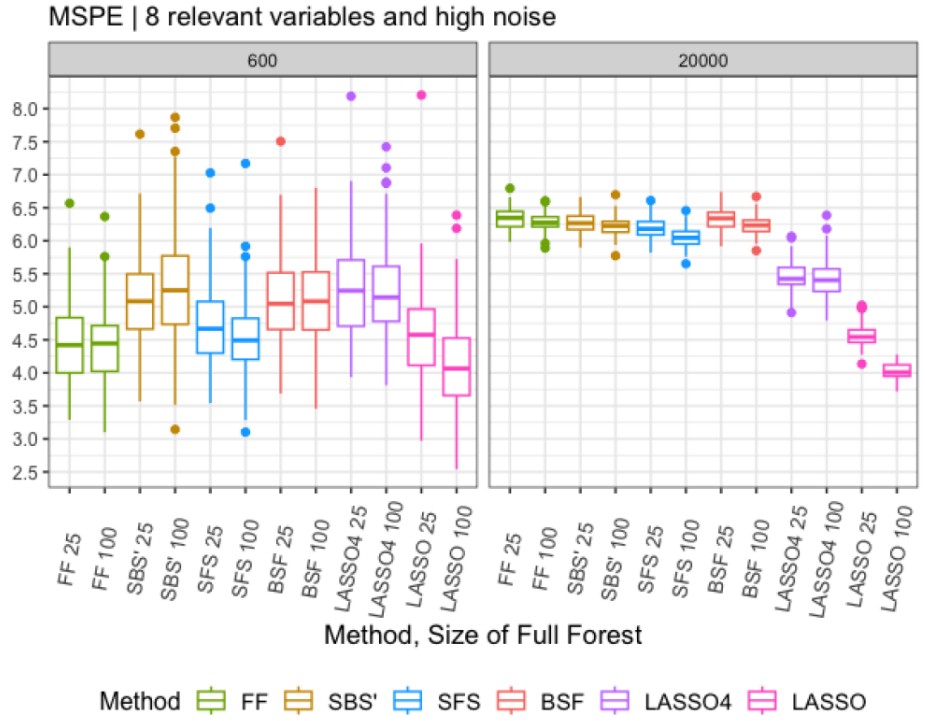

Figure 8: MSPE of full and pruned forests in scenarios with 8 relevant variables and high noise

## G.3. Results with real data

### G.3.1. Iris dataset

150 samples, 4 features (3 continuous, 1 categorical), response: Sepal Length.

Challenging dataset (for forest pruning) due to its very modest sample size.

Table 35: Method performance vs full forest in the Iris dataset

| Method | Avg MSPE | Avg # trees | MSPE$\Delta$ vs FF | p-val | MSPE$\Delta \leq 0$ | Time (s) |
|---|---|---|---|---|---|---|
| Full forest | 0.127 | 24.00 | - | - | - | - |
| SBS' | 0.153 | 2.45 | **+20.5%** | $10^{-12}$ | 18% | 0.01 |
| SFS | 0.137 | 3.87 | **+8.5%** | $10^{-6}$ | 30% | 0.01 |
| BSF | 0.136 | 3.15 | **+7.9%** | $10^{-7}$ | 28% | 0.14 |
| LASSO4 | 0.145 | 3.74 | **+14.8%** | $10^{-12}$ | 20% | 0.48 |
| LASSO | 0.144 | 5.91 | **+13.8%** | $10^{-11}$ | 22% | 0.26 |

Table 36: Direct comparison among methods in the Iris dataset

| Comparison | MSPE$\Delta$ | p-val | MSPE$\Delta \leq 0$ | # trees $\Delta$ | p-val | # trees $\Delta \leq 0$ |
|---|---|---|---|---|---|---|
| SBS' vs SFS | **+11.1%** | $10^{-7}$ | 38% | **-36.7%** | $10^{-8}$ | 90% |
| SBS' vs BSF | **+11.7%** | $10^{-9}$ | 30% | **-22.2%** | $10^{-7}$ | 90% |
| SBS' vs LASSO4 | +5.0% | 0.07 | 44% | **-34.5%** | $10^{-10}$ | 92% |
| SBS' vs LASSO | **+5.9%** | 0.01 | 42% | **-58.4%** | $10^{-13}$ | 95% |
| SFS vs BSF | +0.6% | 0.66 | 59% | **+22.9%** | $10^{-4}$ | 46% |
| SFS vs LASSO4 | **-5.4%** | $10^{-4}$ | 67% | +3.5% | 0.70 | 57% |
| SFS vs LASSO | **-4.6%** | $10^{-3}$ | 65% | **-34.5%** | $10^{-10}$ | 81% |
| BSF vs LASSO4 | **-6.0%** | $10^{-3}$ | 64% | **-15.8%** | $10^{-7}$ | 93% |
| BSF vs LASSO | **-5.2%** | $10^{-3}$ | 64% | **-46.7%** | $10^{-16}$ | 95% |
| LASSO4 vs LASSO | +0.9% | 0.25 | 48% | **-36.7%** | $10^{-14}$ | 97% |

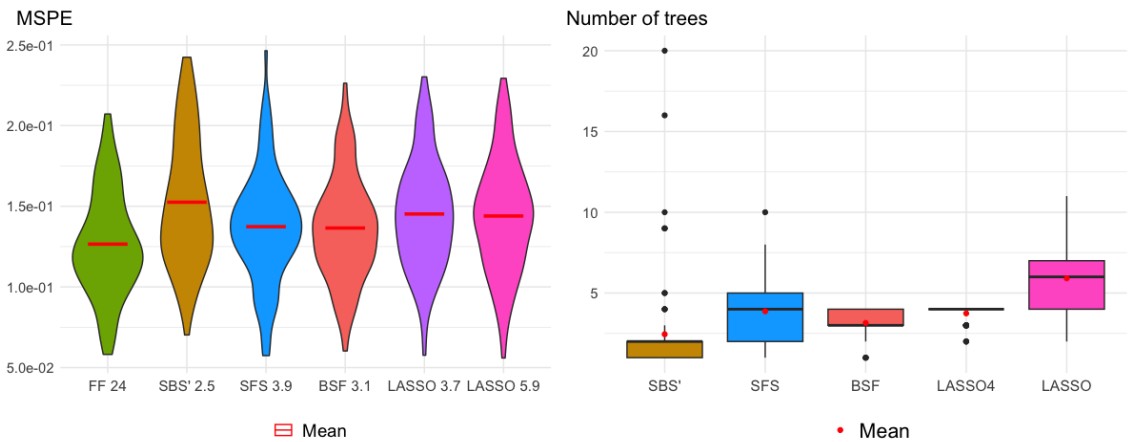

Figure 9: MSPE and number of trees used by different forest-pruning methods in Iris

### G.3.2. Diamonds dataset

21,576 samples, 9 features (6 continuous, 3 ordinal), response: Price.

Relatively good conditions for forest pruning (large sample size).

Table 37: Method performance vs full forest in the Diamonds dataset

| Method | Avg MSPE | Avg # trees | MSPE$\Delta$ vs FF | p-val | MSPE$\Delta \leq 0$ | Time (s) |
|---|---|---|---|---|---|---|
| Full forest | $1.49 \times 10^6$ | 200.00 | - | - | - | - |
| SBS' | $1.22 \times 10^6$ | 2.35 | **-18.0%** | $10^{-16}$ | 99% | 48.14 |
| SFS | $1.11 \times 10^6$ | 10.88 | **-25.6%** | $10^{-16}$ | 100% | 34.78 |
| BSF | $1.14 \times 10^6$ | 2.98 | **-23.3%** | $10^{-16}$ | 100% | 424 |
| LASSO4 | $1.12 \times 10^6$ | 4.00 | **-24.8%** | $10^{-16}$ | 100% | 1.17 |
| LASSO | $1.09 \times 10^6$ | 13.30 | **-26.6%** | $10^{-16}$ | 100% | 0.61 |

Table 38: Direct comparison among methods in the Diamonds dataset

| Comparison | MSPE$\Delta$ | p-val | MSPE$\Delta \leq 0$ | # trees $\Delta$ | p-val | # trees $\Delta \leq 0$ |
|---|---|---|---|---|---|---|
| SBS' vs SFS | **+10.3%** | $10^{-16}$ | 0% | **-78.4%** | $10^{-16}$ | 100% |
| SBS' vs BSF | **+6.9%** | $10^{-16}$ | 7% | **-21.1%** | $10^{-11}$ | 94% |
| SBS' vs LASSO4 | **+9.1%** | $10^{-16}$ | 1% | **-41.3%** | $10^{-16}$ | 96% |
| SBS' vs LASSO | **+11.8%** | $10^{-16}$ | 0% | **-82.3%** | $10^{-16}$ | 100% |
| SFS vs BSF | **-3.0%** | $10^{-16}$ | 95% | **+265%** | $10^{-16}$ | 1% |
| SFS vs LASSO4 | **-1.0%** | $10^{-4}$ | 65% | **+172%** | $10^{-16}$ | 2% |
| SFS vs LASSO | **+1.3%** | $10^{-10}$ | 18% | **-18.2%** | $10^{-9}$ | 83% |
| BSF vs LASSO4 | **+2.1%** | $10^{-12}$ | 17% | **-25.5%** | $10^{-16}$ | 100% |
| BSF vs LASSO | **+4.5%** | $10^{-16}$ | 1% | **-77.6%** | $10^{-16}$ | 100% |
| LASSO4 vs LASSO | **+2.4%** | $10^{-16}$ | 7% | **-69.9%** | $10^{-16}$ | 100% |

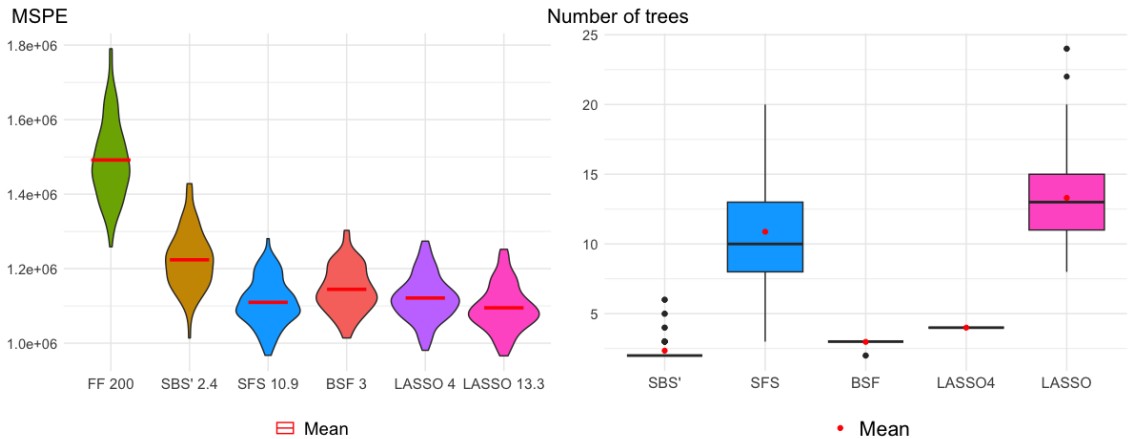

Figure 10: MSPE and number of trees used by different forest-pruning methods in Diamonds

### G.3.3. Midwest dataset

437 samples, 18 selected features (16 continuous, 2 categorical), response: Population.

Challenging dataset due to its low signal-to-noise ratio (SNR) and presence of outliers. If we define the signal-to-noise ratio as $SNR := \frac{R^2}{1-R^2}$ where $R^2$ is the proportion of variance explained by the model we observe that this dataset has one of the lowest SNR among the 19 datasets considered.

Table 39: Method performance vs full forest in the Midwest dataset

| Method | Avg MSPE | Avg # trees | MSPE$\Delta$ vs FF | p-val | MSPE$\Delta \leq 0$ | Time (s) |
|---|---|---|---|---|---|---|
| Full forest | $5.04 \times 10^{10}$ | 200.00 | - | - | - | - |
| SBS' | $5.09 \times 10^{10}$ | 37.04 | +1.1% | 0.09 | 37% | 0.98 |
| SFS | $5.09 \times 10^{10}$ | 15.65 | +1.1% | 0.41 | 49% | 0.86 |
| BSF | $5.28 \times 10^{10}$ | 3.28 | +4.8% | 0.10 | 46% | 203.2 |
| LASSO4 | $6.78 \times 10^{10}$ | 3.56 | **+34.5%** | $10^{-8}$ | 32% | 0.16 |
| LASSO | $6.43 \times 10^{10}$ | 8.41 | **+27.7%** | $10^{-7}$ | 33% | 0.15 |

Table 40: Direct comparison among methods in the Midwest dataset

| Comparison | MSPE$\Delta$ | p-val | MSPE$\Delta \leq 0$ | # trees $\Delta$ | p-val | # trees $\Delta \leq 0$ |
|---|---|---|---|---|---|---|
| SBS' vs SFS | **+0.0%** | 0.04 | 49% | +137% | 0.26 | 73% |
| SBS' vs BSF | -3.5% | 0.61 | 59% | **+1029%** | $10^{-8}$ | 43% |
| SBS' vs LASSO4 | **-24.8%** | $10^{-5}$ | 65% | **+940%** | $10^{-5}$ | 43% |
| SBS' vs LASSO | **-20.8%** | $10^{-4}$ | 61% | +340% | 0.68 | 66% |
| SFS vs BSF | **-3.5%** | $10^{-5}$ | 76% | **+377%** | $10^{-14}$ | 29% |
| SFS vs LASSO4 | **-24.9%** | $10^{-9}$ | 83% | **+340%** | $10^{-13}$ | 28% |
| SFS vs LASSO | **-20.8%** | $10^{-8}$ | 80% | **+86.1%** | $10^{-7}$ | 44% |
| BSF vs LASSO4 | **-22.1%** | $10^{-5}$ | 69% | **-7.9%** | $10^{-3}$ | 94% |
| BSF vs LASSO | **-18.0%** | $10^{-4}$ | 71% | **-61.0%** | $10^{-16}$ | 96% |
| LASSO4 vs LASSO | +5.3% | 0.07 | 44% | **-57.7%** | $10^{-14}$ | 96% |

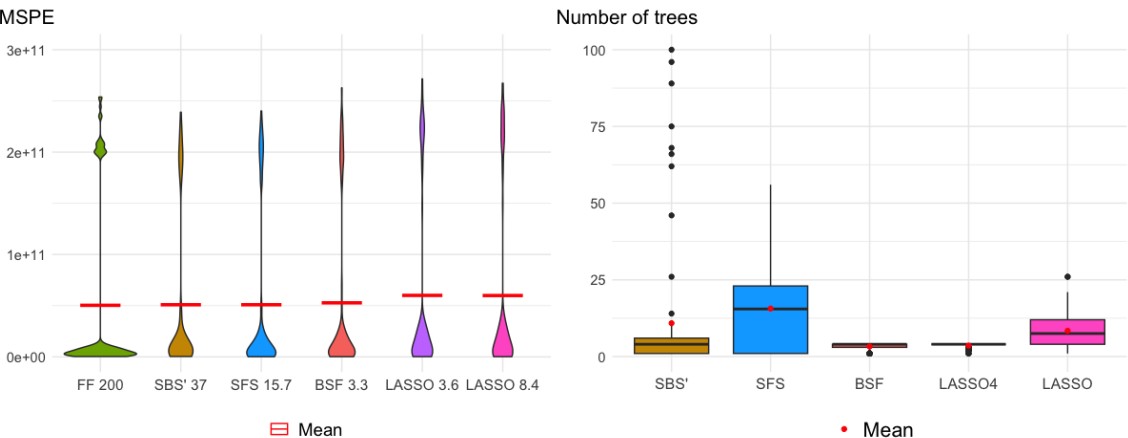

Figure 11: MSPE and number of trees used by different forest-pruning methods in Midwest

# H. Further information on real datasets

## H.1. Iris dataset

This famous (Fisher's or Anderson's) dataset gives the measurements in centimeters of the variables sepal length and width as well as petal length and width, respectively, for 50 flowers from each of 3 species of iris. The species are setosa, versicolor, and virginica. There are 150 cases (rows) and 5 variables (columns) named `Sepal.Length`, `Sepal.Width`, `Petal.Length`, `Petal.Width`, and `Species`. In our test, we predict `Sepal.Length` based on the rest of variables.

| Sepal.Length | Sepal.Width | Petal.Length | Petal.Width | Species |
|---|---|---|---|---|
| 5.1 | 3.5 | 1.4 | 0.2 | setosa |
| 4.9 | 3.0 | 1.4 | 0.2 | setosa |
| 4.7 | 3.2 | 1.3 | 0.2 | setosa |
| 4.6 | 3.1 | 1.5 | 0.2 | setosa |
| 5.0 | 3.6 | 1.4 | 0.2 | setosa |
| 5.4 | 3.9 | 1.7 | 0.4 | setosa |

Figure 12: First 6 rows of the Iris dataset

## H.2. Diamonds dataset

This dataset contains 53940 observations (diamond) and 10 variables: `price`, `carat`, `cut`, `color`, `clarity`, `x` (length), `y` (width), `z` (depth), `depth` percentage $= z/mean(x, y)$, and table (width of top of diamond relative to widest point). We predict `price` as a function of all remaining (6 continuous and 3 ordinal).

| carat | cut | color | clarity | depth | table | price | x | y | z |
|---|---|---|---|---|---|---|---|---|---|
| 0.23 | Ideal | E | SI2 | 61.5 | 55 | 326 | 3.95 | 3.98 | 2.43 |
| 0.21 | Premium | E | SI1 | 59.8 | 61 | 326 | 3.89 | 3.84 | 2.31 |
| 0.23 | Good | E | VS1 | 56.9 | 65 | 327 | 4.05 | 4.07 | 2.31 |
| 0.29 | Premium | I | VS2 | 62.4 | 58 | 334 | 4.20 | 4.23 | 2.63 |
| 0.31 | Good | J | SI2 | 63.3 | 58 | 335 | 4.34 | 4.35 | 2.75 |
| 0.24 | Very Good | J | VVS2 | 62.8 | 57 | 336 | 3.94 | 3.96 | 2.48 |

Figure 13: First 6 rows of the Diamonds dataset

## H.3. Midwest dataset

Demographic information of midwest counties from 2000 US census, containing 437 observations and 28 variables. In our experiment, we predict `poptotal` as a function of the remaining variables after discarding those that are linear combinations of others, leaving us with 19 variables (besides a numeric response, 16 numeric and 2 categorical).

| state | area | poptotal | percwhite | percblack | percamerindan | percasian | percother | popadults | perchsd | percollege | percprof |
|---|---|---|---|---|---|---|---|---|---|---|---|
| IL | 0.052 | 66090 | 96.71206 | 2.5752761 | 0.1482826 | 0.37675897 | 0.18762294 | 43298 | 75.10740 | 19.63139 | 4.355859 |
| IL | 0.014 | 10626 | 66.38434 | 32.9004329 | 0.1788067 | 0.45172219 | 0.08469791 | 6724 | 59.72635 | 11.24331 | 2.870315 |
| IL | 0.022 | 14991 | 96.57128 | 2.8617170 | 0.2334734 | 0.10673071 | 0.22680275 | 9669 | 69.33499 | 17.03382 | 4.488572 |
| IL | 0.017 | 30806 | 95.25417 | 0.4122574 | 0.1493216 | 0.48691813 | 3.69733169 | 19272 | 75.47219 | 17.27895 | 4.197800 |
| IL | 0.018 | 5836 | 90.19877 | 9.3728581 | 0.2398903 | 0.08567512 | 0.10281014 | 3979 | 68.86152 | 14.47600 | 3.367680 |
| IL | 0.050 | 35688 | 98.51210 | 0.1401031 | 0.1821340 | 0.54640215 | 0.61925577 | 23444 | 76.62941 | 18.90462 | 3.275891 |

| poppovertyknown | percpovertyknown | percbelowpoverty | percchildbelowpovert | percadultpoverty | percelderlypoverty | inmetro |
|---|---|---|---|---|---|---|
| 63628 | 96.27478 | 13.151443 | 18.01172 | 11.009776 | 12.443812 | 0 |
| 10529 | 99.08714 | 32.244278 | 45.82651 | 27.385647 | 25.228976 | 0 |
| 14235 | 94.95697 | 12.068844 | 14.03606 | 10.852090 | 12.697410 | 0 |
| 30337 | 98.47757 | 7.209019 | 11.17954 | 5.536013 | 6.217047 | 1 |
| 4815 | 82.50514 | 13.520249 | 13.02289 | 11.143211 | 19.200000 | 0 |
| 35107 | 98.37200 | 10.399635 | 14.15882 | 8.179287 | 11.008586 | 0 |

Figure 14: First 6 rows of the Midwest dataset (selected columns)

# I. Combining regression trees

A regression tree can be represented as a sum of piecewise constant functions over disjoint subsets of the feature space. That is, for some input $x \in \mathbb{R}^d$ a regression tree $t(x)$ is the map $t : \mathbb{R}^d \to \mathbb{R}$, with functional form $t(x) = \sum_{j=1}^M \mathbb{1}(x \in S_j) c_j$ where $\{S_j\}_{j=1}^M$ is a partition of the feature space $\mathbb{R}^d$ into $M$ subsets, and $c_j \in \mathbb{R}$ for $j \in \{1, \ldots, M\}$ is a suitable constant. Therefore, the weighted sum of the images of each tree is itself a tree. As in Quinlan [38] one way to combine two trees into a single tree is to concatenate their architectures: given two trees, $t_1$ and $t_2$, substitute every leaf of $t_1$ by $t_2$ itself, and then set the leaves of the new combined tree to the corresponding weighted sum of the leaves in $t_1$ and $t_2$. This approach ensures that all leaves of each tree are appropriately combined, achieving the desired weighted sum of the images of each tree. Below we illustrate this process with a simple example, combining two small trees with equal weights, where we start with tree $A$ and replace each of its leaves by tree $B$, computing the simple average of the original leaf values in the appropriate leaves of the resulting combined tree. Note that the root's right child in the combined tree has no further children because $x \geq 5 \Rightarrow x \geq 3$, so no further splitting occurs and the predicted response is $(6 + 2)/2 = 4$.

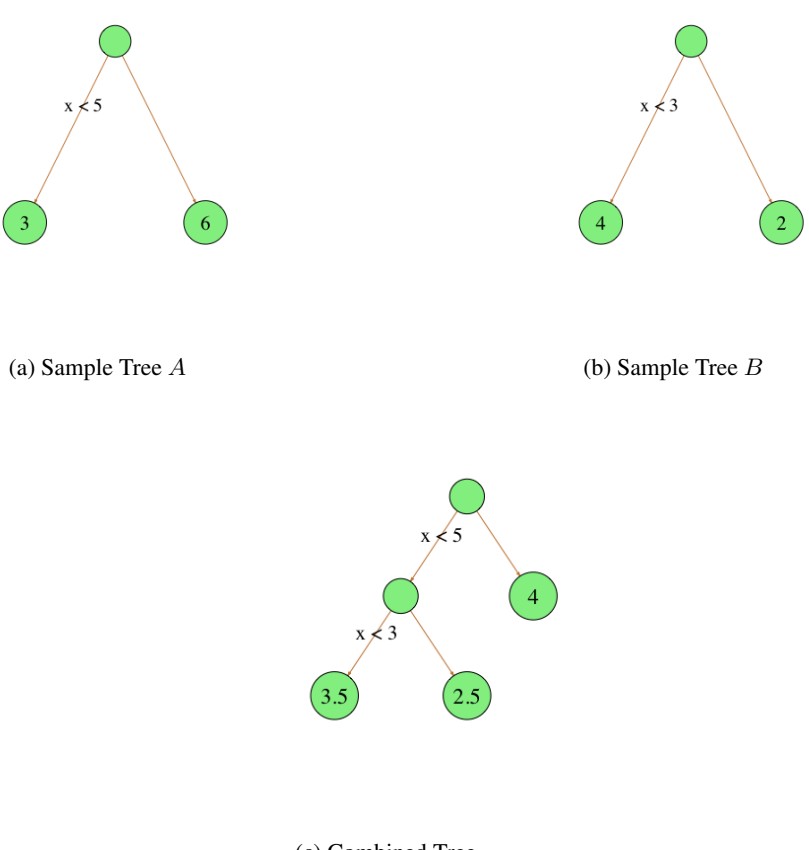

(a) Sample Tree $A$            (b) Sample Tree $B$

(c) Combined Tree

Figure 15: Combining two regression trees (equal weights)

# J. Tree topologies in the Diamonds dataset

In this section we show the structure of both the single pruned tree and the pruned forest that merged the two trees selected by BSF. Although the pruned forest is clearly more complex at a global level, the local complexity is similar for some instances (e.g. small diamonds having carat$< 0.6$ and y$< 5.4$, or large diamonds with carat$> 1.9$ and y$> 7.9$), so individual prediction explanations may not be significantly harder to understand compared to those in the single pruned tree – while MSPE is reduced by $34.2\%$ on average.

To put things in perspective, the number of leaves in a forest like the one in Diamonds with 200 trees, each having at least 5 leaves is of the order of $5^{200} \approx 10^{139}$ which is larger than the number of atoms in the known universe (about $10^{80}$). Therefore, our merged tree, with 49 leaves and better out-of-sample accuracy than the full forest, can be considered a great success.

Indeed, global complexity grows exponentially in the number of trees to be merged, while local complexity grows only linearly. This observation suggests that focusing on local interpretability is a more realistic aspiration, which itself may satisfy the interpretability needs of the relevant stakeholders.

## J.1. Pruned tree

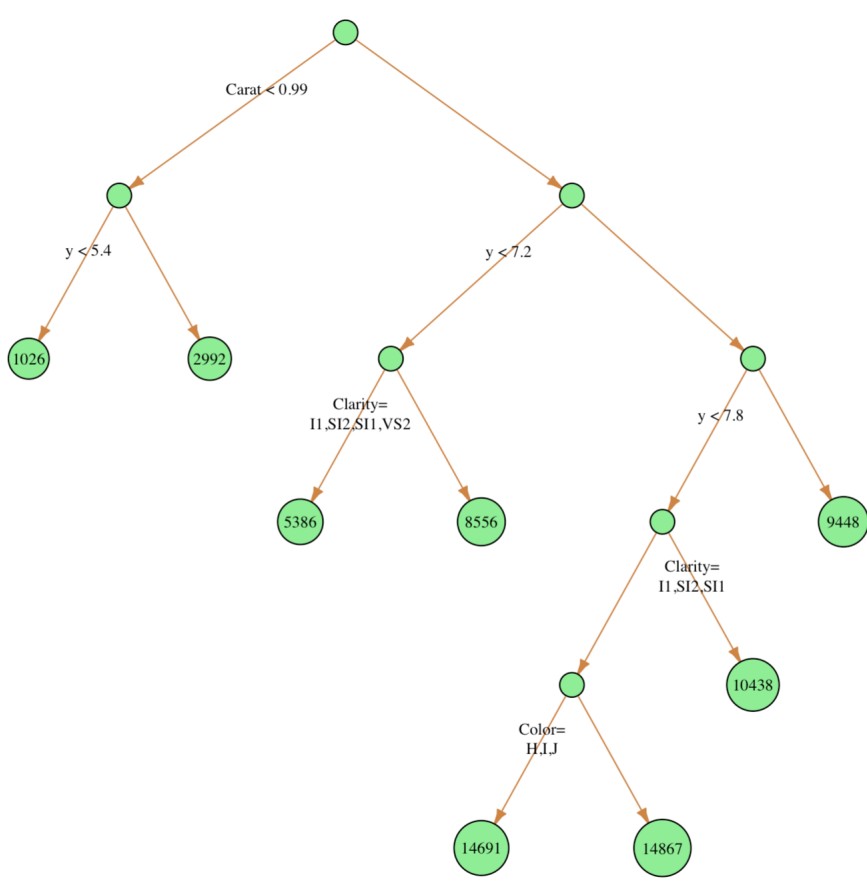

Figure 16: Topology of a pruned tree in the Diamonds dataset

## J.2. Pruned forest

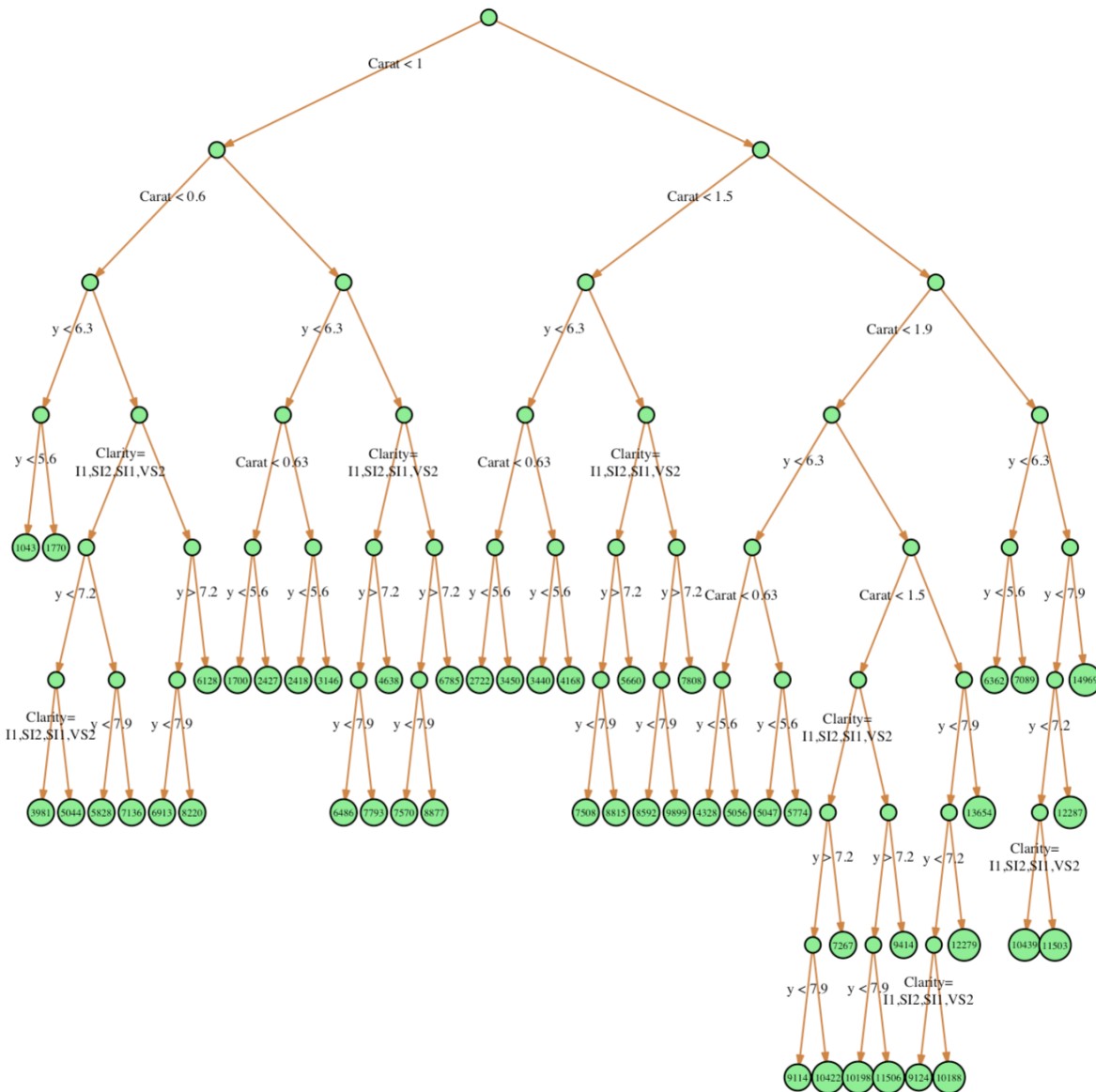

Figure 17: Topology of a BSF-pruned forest in the Diamonds dataset

