# OpenReview forum: "Theoretical and Empirical Advances in Forest Pruning"
_CPAL.cc/2025/Proceedings_Track — CPAL 2025 (Proceedings Track) Poster_

### Official Review · Reviewer_nMSq · 2025-01-07

**Rating:** 7
**Confidence:** 2

**Review:**

**Summary**: This paper studies forest pruning for regression forest. This paper presented theoretical analysis showing that the generalization bound for LASSO-pruned regression forest is no worse than the unpruned forest and then verified this results by experiments.

**Clarity**: This paper is very clear.

**Pro**: The proposed way of pruning forest is natural and amendable to theoretical analysis via Rademacher complexity. The authors proved theoretical results showing that pruning will not hurt the performance of the forest which justifies the pruning method. Further, the experiments showed that the pruning can be very aggressive so that the resulting sub-forest can be merged into a tree which made the model more interpretable.

**Disclaimer**: I am not an expert on regression forest and thus the AC needs to rely on someone who has more expertise in this field.

---

### Official Review · Reviewer_y1y2 · 2025-01-10
**A few clarifying questions on theoretical contributions**

**Rating:** 6
**Confidence:** 4

**Review:**

The submission studies Lasso for pruning random regression forests, i.e. to reduce the number of regression trees used for inference while retaining---or possibly improving---accuracy on a regression task. The submission proposes both theoretical results on the generalization performance of Lasso pruning, and empirical evidence on both synthetic and real-world datasets.

Strengths:
- Pruning and ensembling are important topics
- Experimental results provide evidence in support of the claims in the submission

Weaknesses:
- The setup under which the theoretical results are presented needs clarification
- Presentation is at times hand-wavy

I will expand on my points below and I am looking forward to discussing with the authors to clarify my misunderstandings.

---

1. Which method is the focus of the submission? It is unclear to me whether the submission focuses on BSF or Lasso. The submission first introduces the BSF method, but then mostly discusses Lasso. Could the authors clarify what the focus is?

2. Benefits of non-negative weights. Could the authors expand on the two benefits reported in the main text? I am not sure I understand why the fact that the coefficients may be upper-bounded by 1 in practice leads to improved regularization. Similarly, "variable-selection consistency" and the "Irrepresentable Condition" are used in the text without context, which makes this claim confusing to a first-time reader.

3. Refitting to force a maximum number of trees. Could the authors expand on the need of retraining rather than simply hard-thresholding the coefficients during training to keep the top-k coefficients, where k is the desired number of trees?

Also related to this paragraph, I am not sure I understand the claim about the "expected behavior of $\lambda$". Could the authors expand on what expected behavior they are referring to? Reference [19] makes a different claim from the one reported in the submission. In particular, the submission claims that "the number of nonzero coefficients is not monotonic in the size of $\lambda$" (minor comment, $\lambda$ is a scalar quantity so it does not have a size but a magnitude). This is different from Fig. 5 in reference [19], where, fixed a value of $s$, the number of nonzero coefficients does increase monotonically, but it is stated in the caption that the absolute values of those coefficients may not behave monotonically. Am I misunderstanding the claim made here?

4. Setup of theoretical results. Section 4.1 introduces a linear regression model where the response $Y$ is assumed to be a combination of the outputs of the predictions of a random forest (tangentially, it is somewhat unusual to refer to outputs with $\tilde{X}$ instead of $\hat{Y}$). This seems counterintuitive. Shouldn't the model being studied be defined as a function of the input features, i.e. $Y = \beta^\top X + \epsilon$ (note that Line 158 is missing a transpose sign, $\beta$ and $X$ are multivariate quantities), which the random forest first tries to approximate with $B$ trees, and subsequently claims be made about the generalization performance of the pruned forest with respects to the unknown ground-truth model defined on the covariates? The proposed model in Sec. 4.1 is making a strong assumption about the underlying data-generating process: linearity in the outputs of the random forest. Under this setting, Theorems 4.2, 4.5, and 4.6 are direct applications of existing results on sparse linear classifiers, and their novelty is limited. Could the authors help clarify these misunderstandings?

5. Corollary 4.3. Could the authors expand on how the statement of the theorem supports the claim that "the generalization risk of a Lasso-pruned forest is asymptotically no greater than that of its unpruned counterpart"? If I understand correctly, the equation after Line 181 reads as

$Risk_{pruned} \leq Risk_{unpruned} - (\mathbb{E}(x)^\top(\beta - \beta_U))^2 + C$

but I am not sure I follow how the theorem shows that $C - (\mathbb{E}(x)^\top(\beta - \beta_U))^2$ is always negative. Couldn't this quantity also by positive, which would imply an upper-bound larger than the risk of the unpruned forest?

6. Theorem 4.5. "The distribution $\mathcal{D}$ over $\mathcal{X} \times \mathcal{Y}$" is undefined. Which distribution are the authors referring to? I am not sure I follow the Lipschitz definition included on Lines 199-200. Why is there an assumption of $\|y - y'\| \leq M$? The Lipschitz condition should hold for any $y,y'$? The statement of the theorem uses the terminology of "hypothesis class" coming from learning theory, but this was never used in the submission, and may be confusing to unfamiliar readers. Which hypothesis class is being considered here?

7. Experimental results.

- The synthetic dataset follows the model presented in Sec. 4.1, but the real-world datasets do not. This distinction should be clarified.
- Names of methods should be clarified. What is the difference between "Lasso" and "Lasso4"?
- Why are results from the remaining two real-world datasets not included in the main text?
- Figure 4: why is Lasso not included here?

---

**Minor Comments**
- Lines 82-83: "Given that SFS was shown superior, we test SBS' instead of SBS". I do not follow how the superiority of SFS informs this choice.
- Line 83: SBS' is introduced later on Line 85.
- Line 146: "more accurate than the full forest ...". This claim needs to be supported with experimental results.
- The transition between Sec. 4 and Sec. 5 is abrupt.
- Line 217: "our dataset", which dataset are the authors referring to?
- Line 227: the terms forest and tree induction are used without definition.
- Lines 233-234: "cherry-picking" is informal and unclear in this sentence. What cherry-picking are the authors referring to, and how is it prevented by a fixed random seed? Couldn't the seed be cherry-picked too?
- Figure 2: Missing y-axis label, Missing sub-figure titles.
- Figure 3: the MSPE values are quite high. I assume this is because the dataset is about the price of diamonds. Could the authors include a relative sense of scale of the MSPE?
- Figure 3: if the number after each method's name is the size of the forest, how can Lasso have 13.6 trees?
- Line 291: what are "the images" of trees?
- Lines 298-299: "However, if (local) ... is of consequence". Could the authors clarify this claim? What is local complexity here?
- Line 303: "BSF-pruned forest of just 2 trees" is confusing. It is not actually a forest of two trees being used here, correct? But one tree obtained by merging the BSF forest.

---

### Official Review · Reviewer_4w1z · 2025-01-14

**Rating:** 6
**Confidence:** 3

**Review:**

Strengths:

- The paper provides novel theoretical analysis supporting the empirical success of forest pruning.
- The authors conduct extensive empirical evaluation, testing pruned forests on 16 synthetic scenarios and 3 real datasets
- The paper makes valuable contributions towards enabling interpretability of pruned forests. In some cases, the pruned forest can be reduced to a single merged tree that is much more interpretable than the original forest while preserving accuracy.
- The theoretical results and empirical findings are well-aligned. The theory helps explain the strong performance of the pruning methods observed in the experiments.

Suggestions:

- The theoretical analysis relies on some strong assumptions (e.g. data from residual state space model) and focuses mainly on Lasso pruning. Expanding the scope of the theory by relaxing assumptions and proving results for other pruning methods would further strengthen the contributions.
- While the experiments show no universally best pruning method, the paper could provide more guidance to help practitioners choose methods for a given problem based on dataset characteristics. Some heuristics would be helpful.
- Comparing the pruned forests to additional baselines beyond the full forest, such as other interpretable models like decision trees and sparse linear models, would give more context on the accuracy-interpretability tradeoff achieved by pruning.

---

### Meta-Review · Area_Chair_LbpN · 2025-02-06

**Recommendation:** Accept (Poster)
**Confidence:** 4

**Metareview:**

This paper shows that forest pruning (through Lasso) can improve both accuracy and interpretability in regression. All reviewers have viewed this paper positively and the paper is an excellent fit for CPAL.

---

### Decision · Program_Chairs · 2025-02-11

Accept (Poster)